# Diverse crystal size effects in covalent organic frameworks

Tianqiong Ma [1,2], Lei Wei[3], Lin Liang[2], Shawn Yin[4], Le Xu[1], Jing Niu[2], Huadong Xue[2], Xiaoge Wang[1], Junliang Sun [1✉], Yue-Biao Zhang [3✉] & Wei Wang [2✉]

Crystal size effect is of vital importance in materials science by exerting significant influence on various properties of materials and furthermore their functions. Crystal size effect of covalent organic frameworks (COFs) has never been reported because their controllable synthesis is difficult, despite their promising properties have been exhibited in many aspects. Here, we report the diverse crystal size effects of two representative COFs based on the successful realization of crystal-size-controlled synthesis. For LZU-111 with rigid spiral channels, size effect reflects in pore surface area by influencing the pore integrity, while for flexible COF-300 with straight channels, crystal size controls structural flexibility by altering the number of repeating units, which eventually changes sorption selectivity. With the understanding and insight of the structure-property correlation not only at microscale but also at mesoscale for COFs, this research will push the COF field step forward to a significant advancement in practical applications.

[1] College of Chemistry and Molecular Engineering, Beijing National Laboratory for Molecular Sciences (BNLMS), Peking University, 100871 Beijing, P.R. China. [2] State Key Laboratory of Applied Organic Chemistry, College of Chemistry and Chemical Engineering, Lanzhou University, 730000 Lanzhou, Gansu, P.R. China. [3] School of Physical Science and Technology, ShanghaiTech University, 201210 Shanghai, P.R. China. [4] Drug Product Development Bristol-Myers Squibb Co., One Squibb Drive, New Brunswick, NJ 08903, USA. ✉email: junliang.sun@pku.edu.cn; zhangyb@shanghaitech.edu.cn; wang_wei@lzu.edu.cn

Crystal size represents an important element of control, beyond chemical compositions, for manipulating the physicochemical properties of matter at mesoscale in materials science[1–4]. For example, the crystal size and morphology of zeolites closely relates to the effectiveness of industrial catalysis[5]. For metal-organic frameworks (MOFs), besides having critical impacts on sorption, catalysis, photoelectric property, etc.[6,7], crystal size effect is still one of the most challenging topics in controlling framework flexibility towards smart material[8–17]. As a class of porous crystalline material, covalent organic frameworks (COFs) represents one of the most promising porous materials with a wealth of applications, such as sorption, catalysis, optoelectronics, sensors, drug delivery, etc.[18–25]. Although great advances have been witnessed in the synthesis and utilization of COFs in the past decade, the crystal size of COFs can hardly be controlled over the wide range from nanometre to micrometre because of the crystallization problem of covalent crystals[26,27]. Until recently, we reported the growth of large single-crystal COFs and initially realized the crystal-size-control by a modulation approach[27]. However, further understanding of crystal-size-dependent properties of COFs is still in blank. Here, we successfully apply and optimize our crystal-size-controlled synthetic method to different types of COFs and report the diverse crystal size effects of COFs.

With the control of synthetic precision, LZU-111 and COF-300 are selected as a counter pair in this work to study the crystal size effect (Fig. 1). Although these two COFs are both constructed with tetrahedral nodes (Fig. 1a), they possess different structures with different types of channels and framework motilities. Determined by single-crystal X-ray diffraction (SXRD)[27], LZU-111 crystallizes in the hexagonal system with a threefold interlocked lonsdaleite topology (**lon-b-c3**, Fig. 1b), while COF-300 crystallizes in the tetragonal system with a sevenfold interpenetrated diamond topology (**dia-c7**, Fig. 1c). LZU-111 has three-dimensional (3D) spiral channels and a rigid framework (Fig. 1d) that guest molecules (e.g. $N_2$, 1,4-dioxane) hardly induced its structure transformation[27]. In contrast, COF-300 possesses one-dimensional (1D) straight channels in a flexible framework which can adapt itself upon interacting with guest molecules to form contracted or expanded phases (Fig. 1e)[27–30]. These characteristic structural correlations and differences between two representative COFs motivate us to perform a comparative study with the aim of investigating the diversiform manifestations of crystal size effects in COF systems, which is vital to provide hints for their real applications but still unknown until now.

## Results

**Crystal-size-controllable synthesis.** We previously reported a modulation strategy to grow large single crystals of COFs at room temperature, and obtained different-sized COF-300 crystals by precisely adjusting the amount of modulator[27]. Successfully applying the modulated approach with cooperatively optimizing the synthetic conditions such as freezing, aging, heating, etc. in this work (see details in "Methods" section and Supplementary Figs. 1, 2 and Supplementary Table 1), variable-sized LZU-111 crystals can be controllably synthesized from nanometre to micrometre with narrow size distribution (Fig. 2a–c and Supplementary Figs. 4, 5). For example, 200 nm-sized LZU-111 crystals were synthesized by directly mixing of building units without modulator (Supplementary Fig. 1), resulting in a fast nucleation and crystallization to produce tiny crystals. However, introduction of different amounts of modulator provided varied

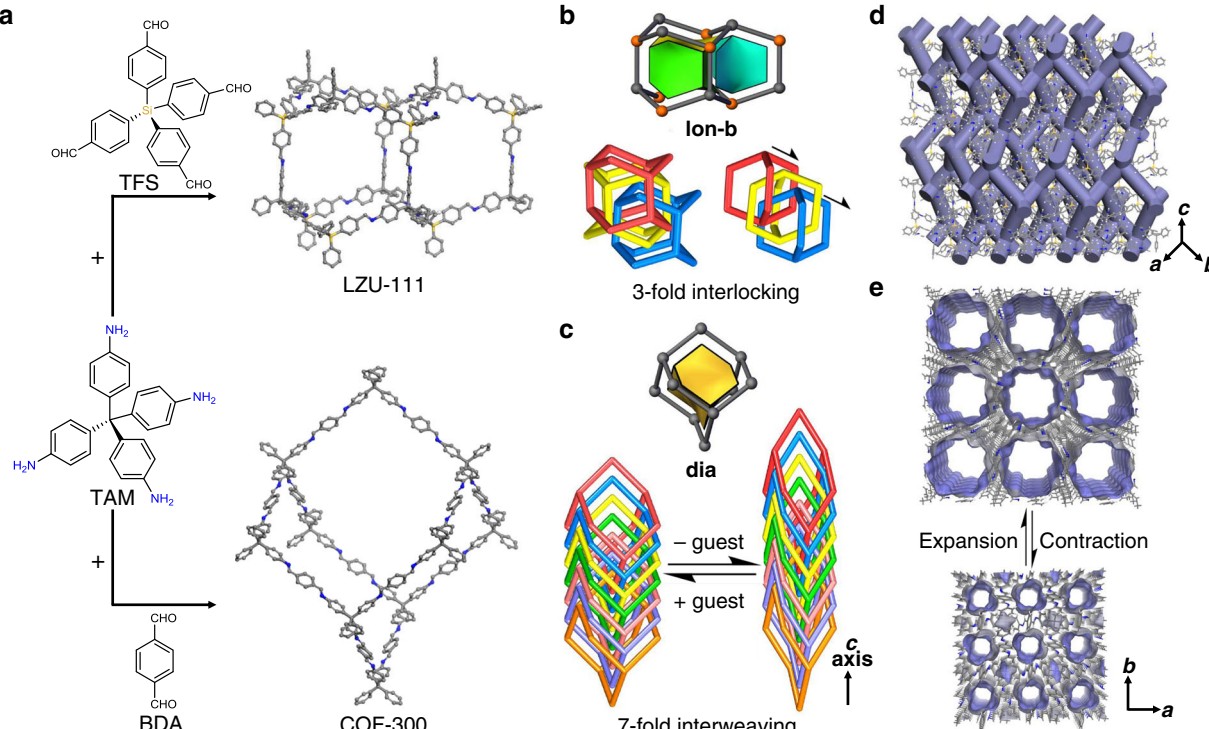

**Fig. 1 Synthetic routes, crystal structures, interpenetration, and porosity diagrams of LZU-111 and COF-300. a** Synthesis and SXRD structures of LZU-111 and COF-300 (C, grey; N, blue; Si, orange. H is omitted for clarity). LZU-111 is connected by tetrakis(4-aminophenyl)methane (TAM) with tetrakis(4-formylphenyl)silane (TFS) while COF-300 is linked by TAM with terephthaldehyde (BDA)[27]. **b** The **lon-b** net adopted by LZU-111 and their interlocked fashion of 3-fold interpenetration. **c** The **dia** net adopted by COF-300 with 7-fold interweaving for structural flexibility upon guest inclusion and removal. **d** The structure with 3D spiral channels of LZU-111. **e** The 1D channel in COF-300 capable for contraction and expansion.

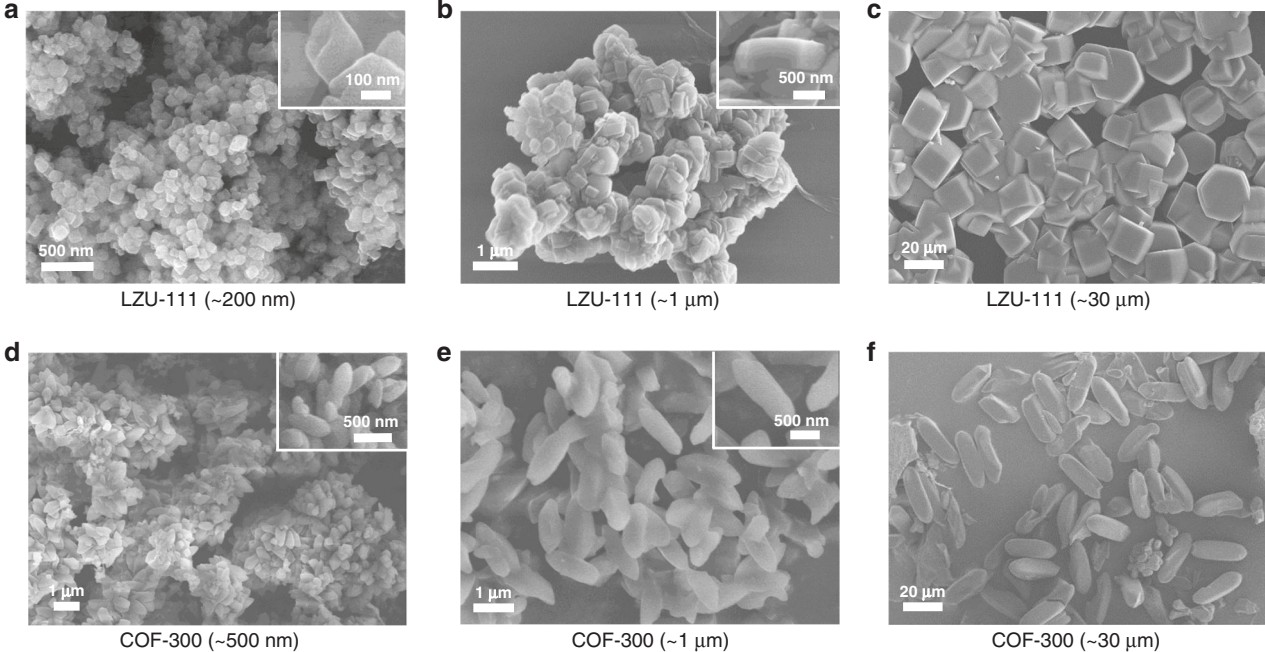

**Fig. 2 Scanning electron microscopy (SEM) images of different-sized LZU-111 and COF-300 crystals. a–c** SEM images of LZU-111 crystals with the average sizes of ~200 nm, ~1, and ~ 30 μm, respectively. **d–f** SEM images of COF-300 crystals with the average sizes of ~500 nm, ~1 and ~30 μm, respectively. For nano-sized and ~1 μm-sized crystals, the magnified images were inserted for clarity.

degrees of nucleation inhibition and different growth controlling forces to slow down the crystallization rate in the formation of different-sized single crystals of LZU-111 (Supplementary Fig. 2). With variable-temperature heating for speeding up the crystal growth, 30 μm-sized LZU-111 crystals suitable for SXRD can be obtained within 10 days, which is greatly shorter than the previous crystal growth time at room temperature (tens of days to reach ~60 μm)[27]. Besides, variable-sized COF-300 crystals (e.g., ~500 nm, ~1 and ~30 μm) were also synthesized (Fig. 2d–f, Supplementary Figs. 3–5 and Supplementary Table 2) for the comparative study.

**Crystal size effects on characterization data.** With the significant variation in the crystal sizes of COFs, the collected characterization data are typically different. For LZU-111, the full width at half maximum (FWHM) of the reflections in powder X-ray diffraction (PXRD) patterns of different-sized samples decreases along with the increase of crystal size from 200 nm to 30 μm (e.g., FWHM values of the strongest reflection peak at 5.6° are 0.338°, 0.221°, 0.211° for 200 nm-sized, 1 and 30 μm-sized LZU-111, respectively, Fig. 3a). It shows a large FWHM difference (0.117°) between 200 nm-sized and 1 μm-sized crystals, while a minor FWHM difference (0.010°) can be observed between 1 and 30 μm-sized LZU-111 crystals. It indicates that the crystallinity of LZU-111 is much improved from the nanocrystals to micrometres crystals. Similarly, all the peaks of solid-state nuclear magnetic resonance (SSNMR) signals of the large single-crystal LZU-111 are much narrower than those of small crystals (Fig. 3b and Supplementary Figs. 6, 7). For example, the line widths of signal at 64 ppm assigned to the quaternary carbon atom in TAM linker are 165, 85, 69 Hz for 200 nm-sized, 1 and 30 μm-sized LZU-111, respectively (Fig. 3b). Furthermore, the resolution of SSNMR signals of larger crystals is improved compared with that of nanocrystals, which can be ascribed to the typically enhanced structure anisotropy of larger crystals in NMR crystallography[31], resulting in two chemically same atoms showing different chemical shifts in SSNMR due to the

crystallographic inequivalence. Besides, there are more defects and mosaics in nanocrystals than in large single-crystalline LZU-111, which is evidenced by the presence of signals of terminal groups on SSNMR (e.g., 190 and 115 ppm, Fig. 3b and Supplementary Fig. 6) and Fourier transform infrared spectroscopy (FT-IR) (Supplementary Fig. 8), and also supported by the mass loss at low temperature in thermogravimetric analysis (Supplementary Fig. 10).

In difference from LZU-111, there exhibits a different mode in corresponding characterization data correlated to the size effect of COF-300. First, the FWHM values of the (200) peak at 8.9° in PXRD patterns of 500 nm-sized, 1 and 30 μm-sized COF-300 are 0.158°, 0.158°, 0.122°, respectively (Fig. 3c). Although the FWHM is further decreased a little in 30 μm-sized COF-300, it should be noted that all different-sized COF-300 crystals keep their single-crystallinity, since the small and large COF-300 crystals were suitable for either single-crystal electron diffraction[28] or SXRD structural analysis[27]. Also there is a relatively small change regarding the linewidth of SSNMR signals in different-sized COF-300. For example, signal at 65 ppm assigned to the quaternary carbon atom in TAM linker for 500 nm-sized, 1 and 30 μm-sized COF-300 are 94, 72, 59 Hz, respectively. Besides, the signals of terminal groups such as aldehyde group on SSNMR and FT-IR spectra are nearly all absent for different-sized COF-300 crystals (Fig. 3d and Supplementary Fig. 9), indicating that different-sized COF-300 crystals analogously contain few defects. Surprisingly, the number of [13]C CP/MAS signals of 30 and 1 μm-sized COF-300 crystals dramatically increases compared with that of 500 nm-sized crystals (Fig. 3d), which implies that the structure anisotropy of large single crystals is much more enhanced in flexible COF-300 compared with that of rigid LZU-111. These results reveal that the characterization data can be analysed and interpreted more reasonably by taking crystal size into consideration. Indeed, such remarkable differences on collected data arising from crystal size effect would not have been observed until realizing the size-controllable synthesis of COFs.

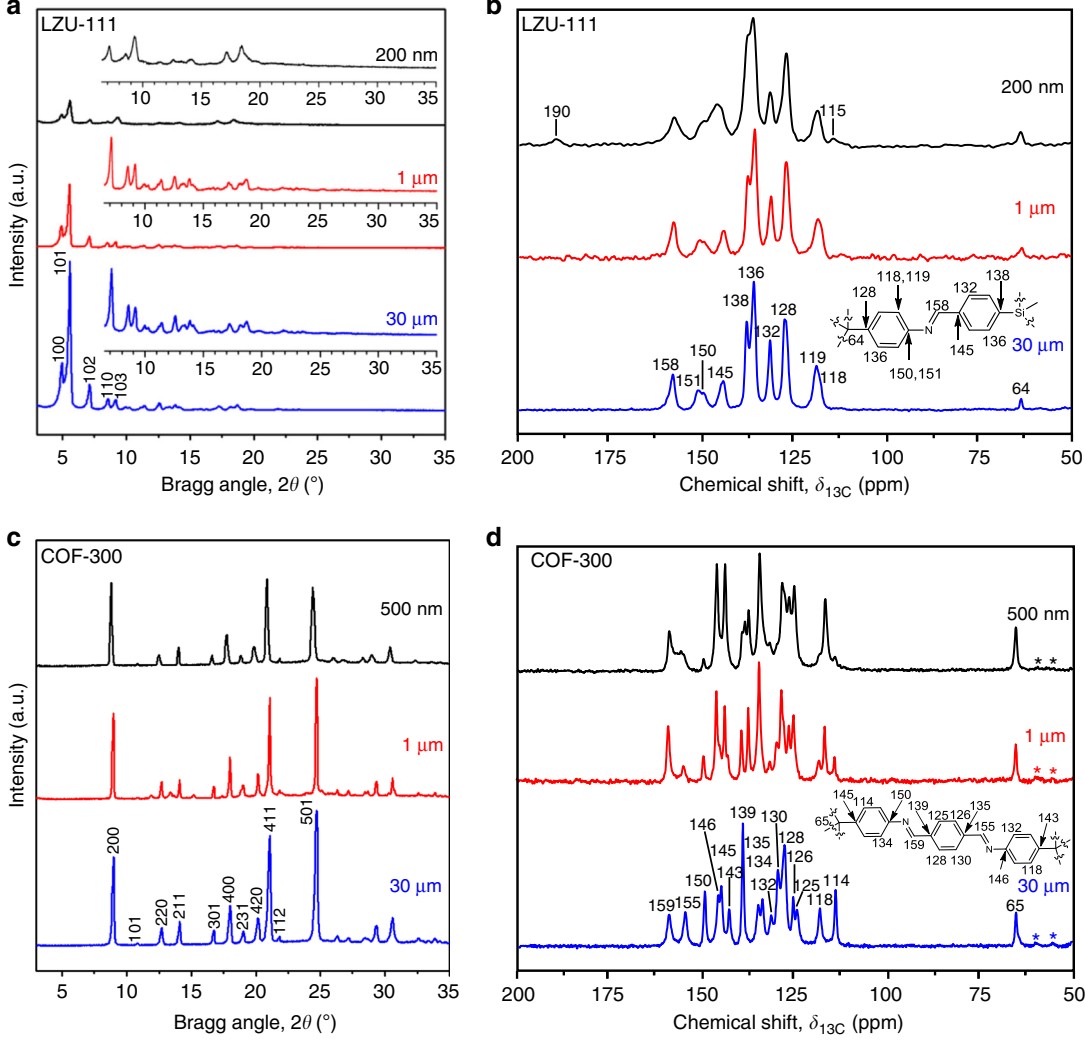

**Fig. 3 PXRD and SSNMR for different-sized LZU-111 and COF-300 (200 nm-sized LZU-111 and 500 nm-sized COF-300, black; 1 μm-sized LZU-111 and COF-300, red; 30 μm-sized LZU-111 and COF-300, blue). a** PXRD patterns of different-sized LZU-111. Inset: magnified patterns of $2\theta = 6-35°$. **b** $^{13}$C CP/ MAS spectra of different-sized LZU-111. The assignment of $^{13}$C chemical shifts are indicated in the chemical structure. **c** PXRD patterns of different-sized COF-300. **d** $^{13}$C CP/MAS spectra of different-sized COF-300. Asterisks denote spinning sidebands. The assignment of $^{13}$C chemical shifts are indicated in the chemical structure.

**Crystal size effects on sorption behaviours**. With the benefit of obtaining COF crystals with significantly different sizes, the influences of crystal sizes on sorption properties of two COFs can be explored. First, the overall $N_2$ uptakes of different-sized LZU-111 are observed to increase along with the increasing crystal size from 200 nm to 30 μm (Fig. 4a). As a result, the Brunauer–Emmet–Teller (BET) surface area of 200 nm-sized LZU-111 is calculated to be 1077 m² g⁻¹, which is much smaller than that of 30 μm-sized single crystals (2120 m² g⁻¹) and theoretical value (2209 m² g⁻¹)[27] (Supplementary Fig. 18). Note that BET surface area of 1 μm-sized LZU-111 is calculated to be 1870 m² g⁻¹, which is much larger than that of 200 nm-sized sample and relatively closer to that of 30 μm-sized crystals. This is in good agreement with the trend of crystallinity change along with crystal size (Fig. 3a). The Ar adsorption–desorption results also display the same variation tendency (Fig. 4b). Secondly, an unconsolidated pore adsorption after $P/P_0 \geq 0.9$ is only observed from the $N_2$ and Ar adsorption isotherms of 200 nm-sized LZU-111 crystals, indicating the pore blockages in poorly crystalline nanocrystals, which is possibly caused by the fast crystallization of nanocrystals in the absence of modulator. The different

regularities of the rigid pore structures in different-sized LZU-111 crystals is further supported by pore size distribution plots (Supplementary Fig. 12) and $^{129}$Xe NMR measurements (Supplementary Fig. 20).

Intriguingly, the crystal size effects on sorption properties of flexible COF-300 are dramatically different from that of rigid LZU-111. Firstly, the overall $N_2$ and Ar uptakes for COF-300 unexpectedly decrease along with the increased crystal sizes (Fig. 4c, d), which is completely opposite to the trend that is observed for LZU-111 (Fig. 4a, b). In fact, the dynamic property of nano-sized COF-300 has been studied in the literature[28–30] and is also reflected here by typical S-shaped $N_2$ and Ar sorption isotherms of 500 nm-sized COF-300. Meanwhile, the contracted and expanded COF-300 structures with different guest molecules ($H_2O$, mother liquor, etc.) in the pores have been confirmed unambiguously by SXRD[27]. However, only very small amount of $N_2$ or Ar molecules can be adsorbed in 30 μm-sized COF-300 single crystals at 77 or 87 K, which is further proved by theoretical calculation (Supplementary Fig. 19) that $N_2$ only fill in the small contracted pores of 30 μm-sized COF-300 and cannot expand the pores. Besides, the $^{129}$Xe NMR measurements confirmed that the

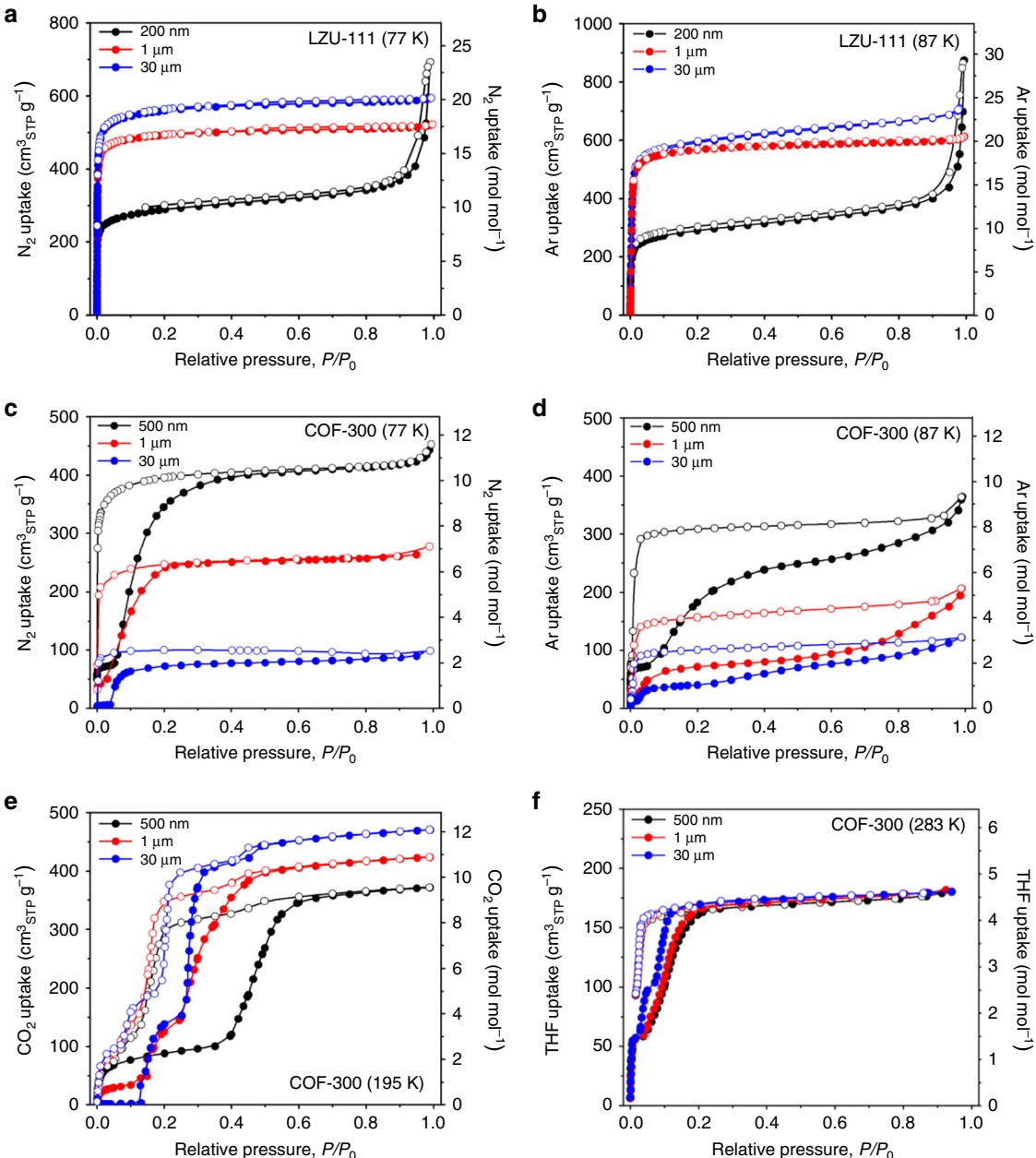

**Fig. 4 Gas sorption experiments and analyses for different-sized COFs (solid circle, adsorption; hollow circle, desorption.** 200 nm-sized LZU-111 and 500 nm-sized COF-300, black; 1 μm-sized LZU-111 and COF-300, red; 30 μm-sized LZU-111 and COF-300, blue). **a**, **b** $N_2$ and Ar adsorption–desorption isotherms of different-sized LZU-111, respectively. **c**-**f** $N_2$, Ar, $CO_2$, and tetrahydrofuran (THF) adsorption–desorption isotherms of different-sized COF-300, respectively.

extents of pore expansion with filling of inert gases in various-sized COF-300 are very different (Supplementary Fig. 21). All the information motivates us to further investigate the discrepant framework flexibility of different-sized COF-300 upon sorption of other different guest molecules. For example, the $CO_2$ sorption experiments for different-sized COF-300 crystals were carried out at 195 K. Interestingly, more $CO_2$ can be adsorbed in large COF-300 crystals than in small crystals (Fig. 4e). This is strikingly different from the inert gas ($N_2$, Ar, and Xe) adsorption behaviour of different-sized COF-300 (Fig. 4c, d and Supplementary Fig. 21). Besides, it is far more intriguing to find that different-sized COF-300 crystals show the same overall uptakes for a specific organic solvent (tetrahydrofuran, 1,4-dioxane, ethyl alcohol, isopropanol) at 298 K (Fig. 4f and Supplementary Figs. 15–17).

## Discussion

With the crystal-size-controlled synthesis, the manufacturing of COFs is more than the regulation of constructional/functional building blocks[32–37] and post-synthetic modification[38] at atomic/molecular level, and also beyond the chemical/physical processing for nano-scaled powder[39–44]. The assembly of COFs now can be precisely achieved at multiple scales[45] from microscopic to mesoscopic level in a bottom-up way. Therefore, the different properties of two COFs with variable crystal sizes are influenced by both intrinsic and extrinsic factors. From the microscopic view, each adamantane-like cage in a subnet of LZU-111 possesses two types of conformations, as shown in Fig. 1b, the chair and the boat forms, driving three lonsdaleite subnets to interlock each other without overlapping by translating to three directions with different offsets. This interpenetration mode prevents the

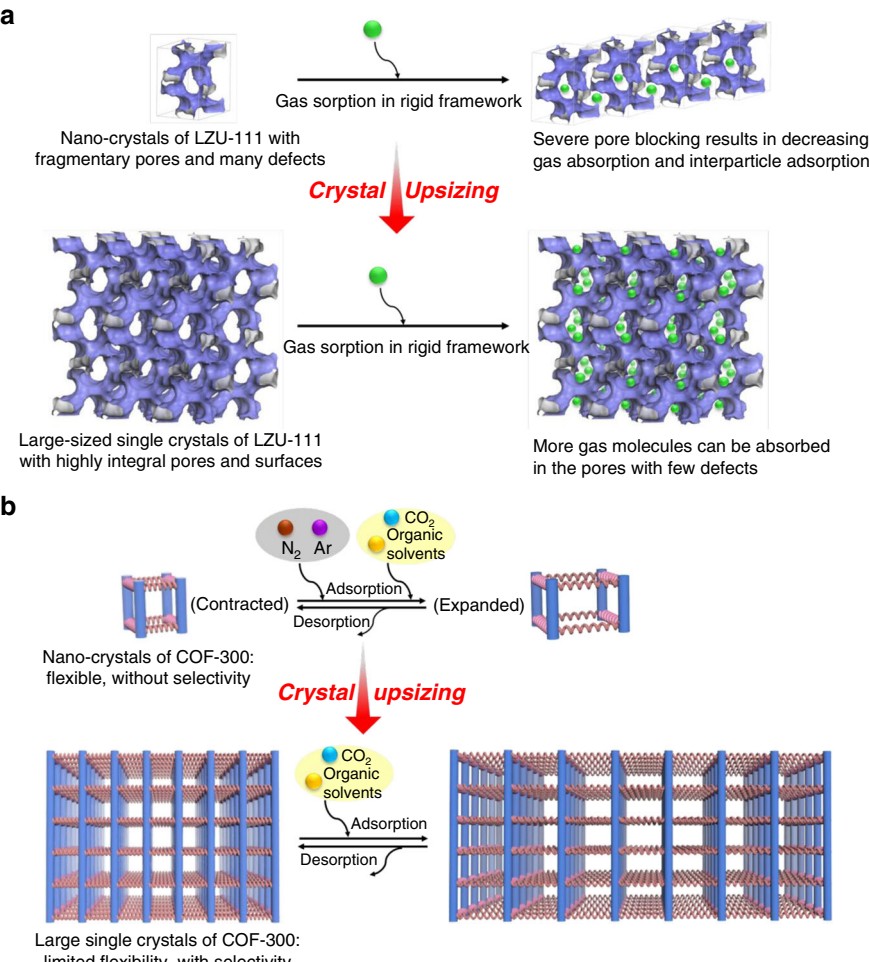

**Fig. 5 Diverse crystal size effects of LZU-111 and COF-300. a** Small crystals of LZU-111 (represented by a 1 × 1 × 1 unit cell) with fragmentary pores and many defects, which has a small gas sorption quantity with some interparticle adsorption. After crystal upsizing, large single crystals of LZU-111 (represented by 3 × 3 × 3 unit cells) adsorb more gas molecules in their highly integral channels. **b** As a springs framework model, small crystals of COF-300 with few repeating units are more flexible, which can absorb different gases without selectivity by changing itself from contracted structure to expanded structure easily. With crystal upsizing, large single crystals of COF-300 with more repeating units are more rigid, selectively adsorbing only $CO_2$ and organic solvents molecules which can interact with the framework to expand pores.

relative sliding between subnets and even limits the deformation of adamantane-like cages, which finally results in a relatively rigid **lon-b-c3** framework of LZU-111 containing spiral channels throughout the framework (Fig. 1d). On the contrary, all the adamantane-like cages in subnets of COF-300 possess solely the chair conformation (Fig. 1c), which is beneficial for seven diamond subnets interweaving to each other along the **c** axis with the same displacement. As a result, the flexible **dia-c7** COF-300 with 1D straight channels formed (Fig. 1e). Upon the adsorption of specific guest molecules, transformation between contracted and expanded structures occurs readily in COF-300 with such a flexible interpenetration mode, since the adamantane-like cages can conform themselves uniformly along **a** and **b** axes without colliding with each other.

With the intrinsic characteristics of both architectures and pore structures, the crystal size at mesoscopic level exerts multiple effects on different COFs, resulting in featured characterization data and diverse adsorption behaviours. With fewer repeating units for small crystals of LZU-111, integrated 3D spiral channels could hardly form and distribute throughout the framework (Fig. 5a). Meanwhile, pore blocking occurs easily within the fragmentary pores containing defects. Consequently, nano-sized

LZU-111 only adsorbed a small amount of gas molecules ($N_2$ and Ar) with an unconsolidated interparticle adsorption, leading to low overall gas uptakes. On the contrary, the 3D channels in larger crystals of LZU-111 with many more unit cells extend to all directions, possessing high pore integrity and less defects, which is evidenced by a series of characterizations (PXRD, SSNMR, $^{129}$Xe NMR, etc.). Therefore, larger crystals of LZU-111 can provide more open space and more accessible sites to adsorb gases, giving rise to much higher gas uptakes.

However, with relatively simple 1D channels extending to only one direction, the channel integrity in different-sized crystals of COF-300 is quite similar in terms of the number of defects, thus the crystallinity of small and large COF-300 crystals is similar (Fig. 3c). However, with the different number of repeating units, different-sized COF-300 show great differences in framework flexibility. Since the larger COF-300 crystals adsorbed fewer inert gases ($N_2$, Ar and Xe, Fig. 4c, d and Supplementary Fig. 21), the structural transformation of COF-300 from contraction to expansion seems much more difficult to be induced in the large crystals than in the nanocrystals by inert gases, indicating that COF-300 framework in larger crystals becomes less flexible. It is reasonable to consider a framework connected by springs as an

analogue model for COF-300 (Fig. 5b) since they both have a specific scalability/flexibility. When there are more springs, the whole structure becomes more rigid and more difficult to be stretched or compressed, while structures with fewer springs are more scalable. However, in difference from the inert gases, $CO_2$ and organic solvents are supposed to interact with COF frameworks[29,46] via weak interactions like N⋯$CO_2$, CH–π or H-bond, which probably induce the structure transformation or phase transition by decreasing energy barrier between the guest-free phase and the guest-accommodated phase. Thus, these molecules can open the pores both in small and large crystals of COF-300 even though the large crystals have higher energy barrier for inert gases. Therefore, the crystal upsizing endows COF-300 with unique sorption selectivity.

Note that there are still several issues which are worthy to be discussed. (1) The gases ($N_2$, Ar and $CO_2$) sorption isotherms of COF-300 (Fig. 4c–e) show that, the larger crystals start to adsorb gas molecules at relatively higher pressures. In detail, 30 μm-sized COF-300 adsorbs almost no gases up to $P/P_0 = 0.05$ for $N_2$ at 77 K, 0.02 for Ar at 87 K, and 0.15 for $CO_2$ at 195 K, which indicates a temperature-dependent, pressure-dependent and gas-dependent adsorption feature[47]. However, 500 nm-sized COF-300 starts adsorbing different gases from very low pressure (e.g., $P/P_0 = 10^{-6}$ for $N_2$). It is probably because a few gas molecules firstly interact with the pore windows of larger COF-300 crystal with longer 1D channels, blocking more molecules from passing into the pores[48], since the framework has no additional open channel along the **a** and **b** axes. This weak blocking is broken later under a certain pressure by a certain amount of gases, then more gases can be adsorbed in the channels. As counterexamples, nano-sized COF-300 crystals with short channels and LZU-111 with 3D channels have no blocking problem, so that nano-sized COF-300 and different-sized LZU-111 start adsorption from very low pressure.

(2) More adsorption/desorption steps could be observed in the both of $CO_2$ and THF isotherms of COF-300 larger crystals. For example, two well-resolved plateaus can be observed in the $CO_2$ isotherm of 500 nm-sized COF-300 (Fig. 4e, black line), which represents a classical flexible-robust sorption process[49]. However, 30 μm-sized COF-300 shows a $CO_2$ adsorption isotherm with multiple steps (Fig. 4e, blue line). Such stepwise sorption behaviour was generally attributed to multiple adsorbent–adsorbate interactions that substantially differ in energy, or various structural transformations occurring under different adsorption pressures[47,50]. Since the short plateaus during gas adsorption represent short equilibrium processes, we assume that the contracted pores in 30 μm-sized COF-300 crystals are opened partially and constantly at certain pressures due to the more rigid framework, until the whole structure is gradually expanded. With unstable host–guest intermediate states, those partially expanded phases can hardly preserve their overall structures, which are reflected in isotherms as ambiguous plateaus. Meanwhile, in the process of overcoming a higher energy barrier to achieve the final structure, more $CO_2$ molecules are adsorbed in larger crystals, resulting in a higher overall $CO_2$ uptake.

(3) Organic solvents (THF, 1,4-dioxane, ethyl alcohol and isopropanol) can also expand the pores of large-sized COF-300 at room temperature, but differing from $CO_2$, the overall uptakes in different-sized COF-300 are approximatively the same. The different host–guest chemistry might be responsible for this diversity, and it is more likely that these guest molecules arrange in the pores specifically inducing by solvent–solvent interaction, which might prevent more solvents adsorbing into the pores. Moreover, different molecular sizes, shapes, and polarities of various solvents also have distinct impacts on the shape of isotherms and their variation tendency (see more details in Supplementary Figs. 15–17 and Supplementary Table 3).

In conclusion, with the controllability at mesoscale in growing COFs crystals with desirable sizes range from hundreds of nanometre to tens of micrometre, we have demonstrated that crystal size had dramatical effect on the crystallinity, structural anisotropy, etc. and further controlled sorption behaviour and the structural flexibility of different COFs. Our data displays that, not only sorption capacity is enhancing with the crystal size increasing in rigid COF, but also sorption selectivity can be regulated by adjusting crystal size in flexible COF. These results imply that crystal size engineering of COFs will offer a promising approach to fabricating high-performance adsorbents and catalysts. It is foreseeable that crystal size effect will also express in photoelectric and sensing COFs and further influence their optical and electrical performances. Therefore, our ability to understand and take advantage of diverse crystal size effect will shed a light on the practical application of COFs in the future.

## Methods

**General procedure for controllable synthesis of LZU-111 crystals in different sizes**. We reported that the adding of aniline as the nucleation inhibitor and competitive modulator not only resulted in the formation of large-sized single crystal COFs, but also provided the possibility for the controllable synthesis of COF crystals with different sizes in our previous work[27]. In this work, the strategy for synthesizing large single-crystal COFs suitable for SXRD can be improved by significantly reducing the reaction time from tens of days at room temperature to 10 days with aging and heating. The crystal size of LZU-111 from 200 nm to 30 μm can well be tuned by adjusting the amount of aniline, with meantime changing the frozen, aging and the heating conditions. The typical procedure is described as follows. A 10 mL glass tube was charged with tetrakis (4-formylphenyl)silane (TFS, 22.4 mg, 0.05 mmol), 1,4-dioxane (0.5 mL), aniline (from 0 to 0.27 mL, 0–60 equiv.), and 0.2 mL of aqueous acetic acid (6.0 M). Tetrakis(4-aminophenyl)methane (TAM, 19.0 mg, 0.05 mmol) was dissolved in 0.5 mL of 1,4-dioxane and then added into the tube. The tube was flash frozen in a liquid $N_2$ bath or in an ice bath, evacuated to vacuum and flame sealed. The fused tube was allowed to stand at ambient temperature for aging within a given time (from 0.5 h to 3 days), and then warmed at 40 °C and heated at 120 °C for a given time (from 1 to 3 days), respectively. Then different-sized LZU-111 crystallized out as light yellow to yellow solids. The crude products were separated from the mother liquid by centrifugation and activated by a series of procedures (such as Soxhlet extracted in 1,4-dioxane for 24 h, dried at ambient temperature for 12 h, and further dried at 120 °C for 12 h) to afford pale yellow solids. The crystal-size-controlled synthesis of COF-300 was reported in our previous work[27,28] and some reaction conditions were optimized accordingly herein. The more detailed information is summarized in Supplementary Tables 1 and 2. The LZU-111 crystals with the average size of ~200 nm, ~1 and ~30 μm, and COF-300 crystals with the average size of ~500 nm, ~1 and ~30 μm were then used for the comparative study, including characterization (such as PXRD, SSNMR, FT-IR, TG, $^{129}$Xe NMR) and gas/vapour sorption experiments ($N_2$, Ar, $CO_2$, organic solvents). The more detailed synthetic procedures and measurement conditions are listed in Supplementary Information.

## Data availability

The data generated and analysed during the current study are included in this published article and its Supplementary Information, which are available from the corresponding author on reasonable request.

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

## Acknowledgements

We thank P.-F. Wei, Y.-T. Chen in Lanzhou University, C. Lin in Peking University and L. Long in ShanghaiTech University for their assistance during gas adsorption measurements. We thank H. Lyu for his assistance on drawing figures. T. Ma thanks for the support of Peking University Boya Postdoctoral Fellowship. The authors gratefully acknowledge the financial support from the National Natural Science Foundation of China (Nos. 21632004, 21527803, 21871009, 21621061 and 21522105).

## Author contributions

T.M., J.S., Y.-B.Z. and W.W. conceived the idea. T.M. synthesized all the COF samples and conducted the characterization and analyses. L.W. and Y.-B.Z. collected the gas/vapour adsorption isotherms of COF-300. L.L. and X.W. took the SEM images. S.Y. and L.X. provided important advises on crystal growth and paper writing. J.N. and H.X. collected the SSNMR spectroscopy. T.M., J.S., Y.-B.Z. and W.W. drafted the manuscript and all the authors commended and revised jointly on it.

## Competing interests

The authors declare no competing interests.
