## [Peer Review File · Nature Communications]

REVIEWER COMMENTS

Reviewer #1 (Remarks to the Author):

The manuscript reports the first case of mesoscopic control of crystal properties in covalent organic frameworks (COFs) based on the crystal size with the state-of-art crystallization methodology. It is interesting to show a significant difference in the dynamic responses of the flexible COF-300, which can be attributed to the equivalently high crystallinity and collectivity of the location motions for diverse crystal sizes. So that, the dynamic response to guest is dependent on the crystal sizes for gate opening. The results have shown cooperativity of the dynamic response from high-quality COF crystals. Although the LZU-111 shows less anomalous behavior due to the contradict inter-locking structure, the authors have also illustrated the new way to quantify crystallinity based on broadening solid-state NMR spectroscopy, PXRD patterns and SEM images. It also ruled out the possibility of different crystallinity of COF-300 with various response. In conclusion, this is an excellent example in soft porous crystals showing crystal size effect on coherently dynamic response. The data quality is of the highest and the presentation is scholarly, and therefore the manuscript could be accepted for publication of Nature Communications after minor revisions. Several points are provided as follow:

1. Replot the scale bar for the SEM images in Fig. 2.
2. Index all the strong peaks in the PXRD patterns and assign the characteristic chemical shifts in Fig. 3. It is very important to highlight the linewidth of SSNMR spectroscopy and the FWHM of PXRD for showing the crystallinity.
3. The crystal size effects on dynamic states of COF-300 are quite interesting. Although Fig. 4 and Fig. 5 provide some conclusions, it is recommended to add a figure or table regarding the relationship of gate-open pressure and sorption capacity, with molecular size, shape and polarity of various gases/vapors.
4. The notation of the net should not be in italic to comply the well-established RCSR code.

Reviewer #2 (Remarks to the Author):

This is an interesting m/s by Junliang Sun, Yue-Biao Zhang, and Wei Wang and co-workers. This m/s has been written well and the logic behind this work is sound. Hence I recommend acceptance of this work per minor revision. This m/s is on the topic of Diverse Crystal Size Effects in Covalent Organic Frameworks. Hence, I believe that this aspect must be explored at least in few other systems than the only imine bonded COFs. For example, Keto enamine bonded COFs as mentioned in ref 23 could be another choice where such phenomenon could be explored. I would say maybe one or two COFs with other types of linkages could prove the diversity of this work.

Since the idea of this work is based on Crystal Size, I would recommend authors to present the DLS data and compare the results based on that. SEM images can only provide a small fraction of the entire product.

Figure 2 and other SEM images are quite blurry. I would recommend authors to look into this aspect.

Overall a nice work and I would recommend acceptance once these aspects are taken care of.

Reviewer #3 (Remarks to the Author):

The authors report the crystal-size controlled synthesis of LZU-111 and COF-300, and investigate the crystal size effect on gas sorption of these two COFs. In one of the authors' previous publications (Ref 27), the crystals of LZU-111 and COF-300 with even larger crystal sizes have been reported and the strategy to control the crystal size was already established, suggesting that the novelty of this work is limited. Although the authors observed (small) changes in PXRD and distinct changes by solid state NMR as well as clearly different gas sorption behaviors with varying crystal sizes for one of the COFs, the understanding of the true origin of these effects is largely phenomenological in nature. In particular, this study is primarily a study on the peculiar sorption behavior of COF-300 (while the overall behavior of LZU-111 is rather standard and predictable). The special sorption behavior of COF-300 is very interesting indeed, but its 1:1 correlation with crystal size effects as invoked by the authors is questionable. While there may well be a correlation, there are numerous other factors that may be at play, which are only indirectly linked to particle size effects. Such effects may include changes in the host-guest interactions by the different amounts of modulator used. This is also what the NMR data suggest, although they are essentially not interpreted here.

Therefore, the originality of this work and the conclusions drawn from it do not meet the high standards of Nat. Commun., which is why I cannot recommend the acceptance of this paper. The authors could consider the following comments to improve the quality of their work:

The authors control the crystal size for LZU-111 and COF-300 by introducing different amounts of aniline modulator. However, the synthesis procedure in the experimental section is rather ambiguous, such as aniline amount "15–60 equiv.", reaction time of "0.5 h–3 d" at room temperature, "1–3 d" at 40 °C, "1–3 d" at 120 °C. The readers cannot acquire a clear information on reaction conditions, including the aniline amount, reaction time at each step, and their influence on crystal size. The ambiguous synthesis also raises questions about the crystal size reproducibility from batch-to-batch. How was this ascertained? The authors should provide statistical SEM information along with complementary characterization techniques such as DLS measurements to provide clear evidence that the given crystal sizes are representative. This is especially important considering that PXRD, solid state NMR, gas sorption and other characterizations requires a large quantity of samples to perform the test, and also because COF-300 500 nm and COF-300 1 μ m have a very small size difference.

The authors intend to claim a different FMHW change trend between LZU-111 and COF-300. Indeed, a noticeable FMHW difference is seen between COF-300 500 nm and COF-300 1 μ m in Figure 3, inconsistent with the claim of "nearly the same" (Page 7). The authors should give the FMHW values for each COF crystal and re-summarize the crystal size effect on PXRD. In addition, the authors investigated the crystal size effect on PXRD with LZU-111 crystals (200 nm, 1 μ m, 30 μ m) and COF-300 crystals (500 nm, 1 μ m, 30 μ m). The crystal sizes found in polycrystalline COF powder may vary substantially, which should also be considered. From Figure 3 a and 3c, an alternative understanding

could be that FMHW shows a large difference with the crystal size below 1 μm , while a minor difference with the crystals larger than 1 μm . The authors should clarify this.

COF-300 1 μm shows extra peaks at around 11°, 13° and 15°, which are absent for COF-300 500 nm and COF-300 30 μm . What is the reason for that?

COF-300 exhibits significant differences in the solid state NMR spectra with varying crystal size, which is a very interesting result. However, it is conceivable that the differences in the ^{13}C NMR spectra are due to varying amounts of residual modulator (or changes in hydration/solvation of the pore system, which has been reported to be a common phenomenon in this system). This is what the authors themselves allude to in the SI, p. 23: “The possible reason is, comparing with 500 nm-sized crystals synthesized without modulator, 1 μm -sized crystals synthesized with aniline has a considerable number of different imine bonds from aniline reacting with aldehyde linkers at the edges of crystals.” without discussing this issue further and its impact on the observed changes in sorption behavior. Unless the origin of the different signals are unambiguously clarified, no meaningful conclusion regarding crystal size effects can be drawn from these data.

In Figure S5, the authors claimed the signals of $-\text{CHO}$ and $-\text{NH}_2$ end groups could be observed for the 200 nm-sized crystals but they are absent for the 30 μm -sized crystals, and related the weaker $-\text{CHO}$ and $-\text{NH}_2$ peak in LZU-111 30 μm with less defects and increased crystallinity. However, LZU-111 1 μm and LZU-111 30 μm show a noticeable intensity at around 114 ppm, which may be due to $-\text{NH}_2$. The authors should provide more NMR spectral evidence to clarify this. Moreover, the fact that the CHO carbon signal is observed in LZU-111 200 nm might be that aniline was not used in the reaction and CHO remained as end group. Instead, a large amount of aniline was used for the synthesis of LZU-111 1 μm and LZU-111 30 μm , which could result in the disappearance of CHO end group carbon signal.

Did the author try crystal-size controlled synthesis for LZU-79, which indeed gave the largest crystal in their previous paper (Ref 27)? As LZU-79 should adopt similar configuration as COF-300, investigating LZU-79 crystal size effect on gas sorption could further verify the generality of the relationship of pore structure and crystal size.

Other comments:

p. 6: “the reflection intensity gradually increases along with the increasing crystal size”: How was the XRD intensity quantified? Unless the amount of material in the X-ray beam is precisely known or an internal standard is used, comparison of X-ray intensities gives qualitative trends at best.

p. 7: It is not clear what the authors mean with the term “structure anisotropy”. Please clarify.

In Figure 3b, 3d and S5, the denotation of asterisks should be described in the figure caption.

Fig S17: “The blue isotherm matches well with the red isotherm, which means that 30 μm -sized single crystals of COF-300 keeps its contracted phase during N_2 adsorption”.

The agreement between the experimental and calculated isotherms is limited, which probably is due to the fact that contraction/expansion of the COF-300 is a dynamic process which is hard to model.

This discrepancy is even more obvious for the black/turquoise isotherms. I therefore encourage the authors to model the sorption behavior by more accurate methods than classical DFT, i.e. using atomistic calculations.

Fig 4: What is the rationale behind the fact that the hysteresis of the smallest sized COF-300 crystals is typically the largest among the different sizes?

The authors should also provide ^{129}Xe NMR measurements for COF-300 and the data should be discussed in the context of the crystal size effects invoked in this manuscript.

Point-by-point Response to The Reviewers' Comments

The comments of each reviewer are copied here in their entirety (*italics*) and our responses are given below for each segment of comments. All the corresponding modifications made in main text and in Supplementary Information were highlighted in yellow.

REVIEWER COMMENTS

Reviewer #1 (Remarks to the Author):

The manuscript reports the first case of mesoscopic control of crystal properties in covalent organic frameworks (COFs) based on the crystal size with the-state-of-art crystallization methodology. It is interesting to show a significant difference in the dynamic responses of the flexible COF-300, which can be attributed to the equivalently high crystallinity and collectivity of the location motions for diverse crystal sizes. So that, the dynamic response to guest is dependent on the crystal sizes for gate opening. The results have shown cooperativity of the dynamic response from high-quality COF crystals. Although the LZU-111 shows less anomalous behavior due to the contradict inter-locking structure, the authors have also illustrated the new way to quantify crystallinity based on broadening solid-state NMR spectroscopy, PXRD patterns and SEM images. It also ruled out the possibility of different crystallinity of COF-300 with various response. In conclusion, this is an excellent example in soft porous crystals showing crystal size effect on coherently dynamic response. The data quality is of the highest and the presentation is scholarly, and therefore the manuscript could be accepted for publication of Nature Communications after minor revisions.

Response: We sincerely appreciate the Reviewer's insightful comments and the support expressed for our work.

Several points are provided as follow:

1. Replot the scale bar for the SEM images in Fig. 2.

Response: As per the Reviewer's suggestion, we replotted the scale bar for the SEM images in revised Fig. 2 (also shown here, see next page).

Fig. 2 | Scanning electron microscopy (SEM) images of different-sized LZU-111 and COF-300 crystals. a-c, SEM images of LZU-111 crystals with the average sizes of ~ 200 nm, $\sim 1 \mu\text{m}$, and $\sim 30 \mu\text{m}$, respectively. **d-f,** SEM images of COF-300 crystals with the average sizes of ~ 500 nm, $\sim 1 \mu\text{m}$, and $\sim 30 \mu\text{m}$, respectively. For nano- and $\sim 1 \mu\text{m}$ -sized crystals, the magnified images were inserted for clarity.

2. Index all the strong peaks in the PXRD patterns and assign the characteristic chemical shifts in Fig. 3. It is very important to highlight the line width of SSNMR spectroscopy and the FWHM of PXRD line width of for showing the crystallinity.

Response: According to the Reviewer’s helpful suggestion, we marked the index of all the strong peaks in the PXRD patterns and assigned characteristic chemical shifts in SSNMR spectra in revised Fig. 3 (also as shown below). “The assignment of ^{13}C chemical shifts are indicated in the chemical structure” was added accordingly in the legend of Fig. 3. Besides, the corresponding description of line width of SSNMR spectroscopy and the FWHM of PXRD peaks have been added in the revised main text to highlight the variation tendency of crystallinity. They are listed as follows:

Paragraph 4 on Page 6: “...e.g., FWHM values of the strongest reflection peak at 5.6° are 0.338° , 0.221° , 0.211° for 200 nm-, $1 \mu\text{m}$ - and $30 \mu\text{m}$ -sized LZU-111, respectively, ...”

Paragraph 4 on Page 7: “For example, the line widths of signal at 64 ppm assigned to the quaternary carbon atom in TAM linker are 165 Hz, 85 Hz, 69 Hz for 200 nm-, $1 \mu\text{m}$ - and $30 \mu\text{m}$ -sized LZU-111, respectively (Fig. 3b).”

Paragraph 5 on Page 7: “...the FWHM values of the (200) peak at 8.9° in PXRD patterns of 500 nm-, $1 \mu\text{m}$ - and $30 \mu\text{m}$ -sized COF-300 are 0.158° , 0.158° , 0.122° , respectively (Fig. 3c).”

Paragraph 5 on Page 8: “There is a relatively small change regarding the linewidth of SSNMR signals in different sized COF-300. For example, signal at 65 ppm assigned to the quaternary carbon atom in TAM linker for 500 nm-, $1 \mu\text{m}$ - and $30 \mu\text{m}$ -sized COF-300 are 94 Hz, 72 Hz, 59 Hz, respectively.”

Fig. 3 | PXRD and SSNMR spectra for different-sized LZU-111 and COF-300. **a**, PXRD patterns of different-sized LZU-111. Inset: magnified patterns of $2\theta = 6 - 35^\circ$ with normalized intensity. **b**, ^{13}C CP/MAS spectra of different-sized LZU-111. The assignments of ^{13}C chemical shifts are indicated in the chemical structure. **c**, PXRD patterns of different-sized COF-300. **d**, ^{13}C CP/MAS spectra of different-sized COF-300. Asterisks denote spinning sidebands. The assignments of ^{13}C chemical shifts are indicated in the chemical structure.

3. The crystal size effects on dynamic states of COF-300 are quite interesting. Although Fig. 4 and Fig. 5 provide some conclusions, it is recommended to add a figure or table regarding the relationship of gate-open pressure and sorption capacity, with molecular size, shape and polarity of various gases/vapors.

Response: We thank the Reviewer very much for being interested in our results. As suggested by Reviewer, we added a table (Table S3) in Supplementary Information for showing the relationship between gate-open pressure and sorption capacity of COF-300 with different guest molecular size, shape and polarity of various gases/vapors.

Table S3. Relationship between gases/vapors properties and sorption behaviors of different sized COF-300.

Gas/ vapor	Molecular size and shape	Polarity comparison	Gate-open pressure (P/P_0)			Sorption capacity ($\text{cm}^3_{\text{STP}} \text{g}^{-1}$)		
			500 nm	1 μm	30 μm	500 nm	1 μm	30 μm
Ar 	sphere	Ar < N ₂ < CO ₂	0.05	0.025 ^a	NA ^b	364.4	206.8	122.5
N ₂ 	2.99 Å ellipsoid		0.05	0.028 ^a	NA ^b	436.9	274.4	115.2
CO ₂ 	linear		0.49	0.149	0.129	372.5	423.9	426.4
Tetrahydrofuran 	4.053 Å ring molecule	THF < 1,4-dioxane < isopropanol < ethanol	0.038	0.039	0.015	179.0	181.8	180.4
1,4-dioxane 	4.796 Å ring molecule		0.055	0.062	0.027	171.7	174.2	171.9
Isopropanol 	4.322 Å rod-like		0.011	0.091	0.085	178.2	179.3	180.5
Ethanol 	4.784 Å rod-like		0.273	0.276	0.256	219.5	221.1	223.5

a. The pores were partially opened in 1 μm -sized COF-300 with N₂ and Ar. *b.* The pores can hardly be opened in 30 μm -sized COF-300 with N₂ and Ar, see Fig. S19.

4. *The notation of the net should not be in italic to comply the well-established RCSR code.*

Response: At the suggestion of the Reviewer, we replaced all the italic notation of RCSR codes as non-italic in the revised manuscript.

Reviewer #2 (Remarks to the Author):

This is an interesting m/s by Junliang Sun, Yue-Biao Zhang, and Wei Wang and co-workers. This m/s has been written well and the logic behind this work is sound. Hence I recommend acceptance of this work per minor revision.

This m/s is on the topic of Diverse Crystal Size Effects in Covalent Organic Frameworks. Hence, I believe that this aspect must be explored at least in few other systems than the only imine bonded COFs. For example, Keto enamine bonded COFs as mentioned in ref 23 could be another choice where such phenomenon could be explored. I would say maybe one or two COFs with other types of linkages could prove the diversity of this work. Since the idea of this work is based on Crystal Size, I would recommend authors to present the DLS data and compare the results based on that. SEM images can only provide a small fraction of the entire product. Figure 2 and other SEM images are quite blurry. I would recommend authors to look into this aspect. Overall a nice work and I would recommend acceptance once these aspects are taken care of.

Response: We deeply appreciate the Reviewer's valuable comments and the recommendation for our work.

1. It is really true as Reviewer pointed out that “*this aspect must be explored at least in few other systems than the only imine bonded COFs*”. **First**, the generality of this kind of modulated method we used was also reported in B-O linked COFs to increase the crystallinity of crystallization domain [*J. Am. Chem. Soc.* 2014, *136*, 8783-8789; *J. Am. Chem. Soc.* 2015, *138*, 1234–1239]. Unfortunately, all these works were failed to obtain large single crystals which were suitable for single-crystal x-ray diffraction. **Second**, as we know that growing of COF single crystals is extremely difficult since its strong covalent linkage and low reversibility however contradict to error-check process in crystal growth. We reported the first large single crystals and their single-crystal x-ray diffraction structure [*Science* 2018, *361*, 48–52], 13 years after the first COF was reported in 2005. After that, we continued to explore the generality of the synthetic strategy, which was also applied in our recent published work [*Angew. Chem. Int. Ed.* 2020, DOI: 10.1002/anie.202007230]. In this manuscript, the original synthetic method was extended to the controllably synthesis of different-sized COF crystals, which was the development of another aspect of COF crystal growth. **Third**, although controllable synthesis or single-crystal growth of different COFs with other linkage is not easy, we are dedicating ourselves to growing covalent crystals/COFs with other types of linkages, which are still under investigation and will provide new knowledge of the COF dynamic functions in our future publications. We fully agree with Reviewer's comment that Keto enamine bonded COFs would also be one of excellent candidates. We believe this is a promising project regarding crystal engineering because crystallization is always a complicated and full of unpredictable magic.

2. As correctly suggested by the Reviewer, DLS measurement is usually used to determine the particle size distribution of crystals in suspension or in solution. We tried our best to measure particle size distribution by DLS in order to compare the size distribution with SEM imaging results. Unfortunately, after screening different measurement conditions including suspension solvents, ultrasonic time during sample preparation, and different concentration of disperse system, etc., we found these COF samples were not suitable for DLS measurements for the following reasons.

1) From SEM images, we can see COF samples aggregated severely, even if the samples were treated after ultrasonication. **2)** If the samples were sonicated for long time and under high powder, parts of the crystals were broken into pieces with different sizes (Fig. R1), resulting a polydispersity in the measurement. **3)** There were

macroscopic sedimenting particles in the DLS measurements when crystal size of COFs is larger than $5\ \mu\text{m}$. **4)** COF-300 can be regarded as a material with swelling properties [*Science* 2018, 361, 48–52; *J. Am. Chem. Soc.* 2019, 141, 3298–3303], which is not suitable for DLS measurement. Specifically, with different guest molecules or solvents, the structure of flexible COF-300 transformed between contract and expanded phases. The unit cell parameter changed from $a = b = 19.6394(9)\ \text{\AA}$, $c = 8.9062(4)\ \text{\AA}$ to $a = b = 26.2260(18)\ \text{\AA}$, $c = 7.5743(10)\ \text{\AA}$, with the unit-cell volume changed from $3435.2(4)\ \text{\AA}^3$ to $5209.6(10)\ \text{\AA}^3$. This significant increase in unit cell volume led to an obvious change on crystal size and morphology. As shown in Fig. R2, COF-300 crystals in water have rod-like morphology with crystal size of $55\ \mu\text{m} \times 10\ \mu\text{m}$, After exchanged with 1,4-dioxane, the same batch of sample turned into bar-like morphology with crystal size of $45\ \mu\text{m} \times 15\ \mu\text{m}$ [also see in our previous work: *Science* 2018, 361, 48–52]. Hence, there would be no comparison between crystal sizes measured in solvent (DLS) and those under vacuum (SEM). **5)** Although water would not cause swelling of COF-300, it is not a good dispersant for organic hydrophobic COF material.

Fig. R1. Comparison of SEM images of $1\ \mu\text{m}$ COF-300 crystals before and after sonication. It shows that after sonication over 15 mins, the crystals were broken into pieces with different sizes.

Fig. R2. Optical microscope photographs of COF-300 in water and in dioxane.

To clarify the average crystal size of LZU-111 and COF-300, we would like to show more low-magnification SEM images of LZU-111 samples from different batches and COF-300 samples prepared in different labs (Lanzhou University and Peking University), which can help to provide more size distribution information for samples. See Table S1 and S2. Besides, we carefully analyzed the SEM information for each COF material in Fig. 2 and provided the statistical crystal size distribution as Fig. S4 in revised Supplementary Information. All these data showed that the synthesis of the LZU-111 and COF-300 crystals are well-reproduced, and each sample possess high uniformity with very narrow size distribution.

Table S1. Summary for controllable synthesis of different-sized LZU-111.

LZU-111							
Size	Aniline	Frozen	Rt Aging	Warming (40 °C)	Heating (120 °C)	Batch 1	Batch 2
~200 nm	—	liquid N ₂	0.5 h	—	3d		~1 μm ^a	15 eq, 0.07 mL	Liquid N ₂	1d	3d	3d		~30 μm	60 eq, 0.27 mL	Ice bath	3d	3d	4d		
a. The crystal size distributed from 700 nm to 1 μm, represented by ~1 μm-sized LZU-111.

Table S2. Summary for controllable synthesis of different-sized and COF-300.

COF-300 ^a							
Size	Aniline	Frozen	Rt Aging	Warming (40 °C)	Heating (120 °C)	Sample synthesized in Lanzhou University	Sample synthesized in Peking University
~500 nm ^b	–	Liquid N ₂	0.5 h	–	3d		
~1 μm	0.6 eq, 5 μL	Liquid N ₂	1d	3d	3d		
~30 μm	15 eq, 0.12 mL	Ice bath	3d	3d	4d		

a. The samples synthesized in Lanzhou University (LZU) were operated in sealed glass tubes, and samples synthesized in Peking University (PKU) were operated in sealed pressure tubes. *b.* These nanocrystals were synthesized with 15 M HOAc⁵.

Fig. S4 | Statistical crystal size distribution (corresponding to SEM information in Fig. 2 in main text).

The statistics was implemented and analyzed with *Image J*. **a**, The size of nano LZU-111 crystal distributed from 150 to 270 nm and centered at 200 nm, represented by 200 nm-sized LZU-111. **b**, The size of LZU-111 crystal distributed mainly from 0.7 to 1.0 μm, represented by 1 μm -sized LZU-111. **c**, The size of micro-sized LZU-111 crystal distributed from 24 to 31 μm, represented by 30 μm-sized LZU-111. **d**, The size of nano COF-300 crystal mainly distributed from 450 to 600 nm and centered at 500 nm, represented by 500 nm-sized COF-300. **e**, The size of COF-300 crystal distributed mainly from 0.8 to 1.2 μm and centered at 1.0 μm, represented by 1 μm-sized COF-300. **f**, The size of micro-sized COF-300 crystal distributed from 24 to 33 μm, represented by 30 μm-sized COF-300.

3. Considering the Reviewer's suggestion, we have replaced some of the SEM images in Fig. 2 with new clear images to show the well-defined morphology and narrow size distribution.

Fig. 2 | Scanning electron microscopy (SEM) images of different-sized LZU-111 and COF-300 crystals. a-c, SEM images of LZU-111 crystals with the average sizes of ~200 nm, ~1 μm, and ~30 μm, respectively. **d-f,** SEM images of COF-300 crystals with the average sizes of ~500 nm, ~1 μm, and ~30 μm, respectively. For nano- and ~1 μm-sized crystals, the magnified images were inserted for clarity.

Reviewer #3 (Remarks to the Author):

The authors report the crystal-size controlled synthesis of LZU-111 and COF-300, and investigate the crystal size effect on gas sorption of these two COFs. In one of the authors' previous publications (Ref 27), the crystals of LZU-111 and COF-300 with even larger crystal sizes have been reported and the strategy to control the crystal size was already established, suggesting that the novelty of this work is limited. Although the authors observed (small) changes in PXRD and distinct changes by solid state NMR as well as clearly different gas sorption behaviors with varying crystal sizes for one of the COFs, the understanding of the true origin of these effects is largely phenomenological in nature. In particular, this study is primarily a study on the peculiar sorption behavior of COF-300 (while the overall behavior of LZU-111 is rather standard and predictable). The special sorption behavior of COF-300 is very interesting indeed, but its 1:1 correlation with crystal size effects as invoked by the authors is questionable. While there may well be a correlation, there are numerous other factors that may be at play, which are only indirectly linked to particle size effects. Such effects may include changes in the host-guest interactions by the different amounts of modulator used. This is also what the NMR data suggest, although they are essentially not interpreted here. Therefore, the originality of this work and the conclusions drawn from it do not meet the high standards of Nat. Commun., which is why I cannot recommend the acceptance of this paper.

Response: We thank the Reviewer for the valuable comments on our work. Here, we hope to take this opportunity to present the novelty and importance of this work as following:

- 1) We reported in this manuscript the first case of mesoscopic control of properties and functions in COFs with crystal size ranging from hundreds of nanometres to tens of micrometres. It is not only a significant issue because COFs is becoming an important porous material in many fields including sorption, catalysis, etc., but also an innovative work since the relationship between COF crystal size and its property has never been reported yet. All these studies were based on the realization of crystal-size-controlled synthesis. We have already pointed out in the manuscript that this work drew the essence of our previously reported synthetic strategy for growing large single crystals, but further optimized the method: The reaction time of obtaining large crystals which is suitable for SXRD is greatly decreased from tens of days to several days, which is a significant improvement for such time-consuming synthesis. Besides, the generality of size-controlled synthesis was also demonstrated here in another COF (LZU-111). Regarding the innovation of this work, materials property and function study that closely relating to crystal size effect is our key point, and many important manipulation can be realized such as largely increasing the surface area of porous material, framework flexibility control, etc., which have not been reported in COF field to our best knowledge.
- 2) We really agree with Reviewer that the flexibility control of COF-300 by regulating crystal size effect is quite fantastic and has never been observed in COFs, and this topic is also very challenging in flexible MOFs [Science 2013, 339, 193–196]. In this work, we would like to explore more by choosing two COF examples (LZU-111 vs. COF-300) as a perfect counter pair, regarding their similarities and differences. For example, **lon** topology vs. **dia** topology, spiral channels vs. straight channels, rigid framework vs. flexible framework. This comparative study can not only provide new insights of different COF property in sorption, but also unveil the structure-property relationship persuasively. As correctly pointed by Reviewer, “*there are numerous other factors that may be at play*”, this is exactly what we want to study in our manuscript: how these factors (crystallinity, topology, channel direction, channel shape, framework flexibility, and also trace of residue/terminal groups in some cases etc.) acted on the sorption behaviors. Hence, we would like to do

a comparison study to discuss deeply about the structure-property relationship instead of only presenting sorption property of COF-300 as a single example.

- 3) The role of the modulator used in the synthetic reactions is to increase reversibility of the whole crystal growth process. It has already been proved by atomically accurate SXRD technique that residual aniline can hardly be detected in single-crystal COFs [*Science* 2018, 361, 48–52]. We would like to provide more evidences in the response to the following questions to support our conclusion.

The authors could consider the following comments to improve the quality of their work: The authors control the crystal size for LZU-111 and COF-300 by introducing different amounts of aniline modulator. However, the synthesis procedure in the experimental section is rather ambiguous, such as aniline amount “15–60 equiv.”, reaction time of “0.5 h–3 d” at room temperature, “1–3 d” at 40 °C, “1–3 d” at 120 °C. The readers cannot acquire a clear information on reaction conditions, including the aniline amount, reaction time at each step, and their influence on crystal size. The ambiguous synthesis also raises questions about the crystal size reproducibility from batch-to-batch. How was this ascertained?

Response: Thank you for the Reviewer’s helpful suggestions. To clarify clearly the corresponding synthesis conditions for different COF materials, we added two detailed tables (Tables S1-S2, also shown as following) in the Supplementary Information after the conclusive description of synthetic method, which gave all the specific conditions and corresponding SEM images for different samples. Accordingly, this information was also added in the Method part in main text: “The more detailed information was summarized in Tables S1-S2.”

To demonstrate the generality and reproducibility of the synthesis, the SEM images of LZU-111 samples from other different batches are shown in Table S1 and two more batches of COF-300 samples that synthesized in different labs from Lanzhou University (LZU, in Lanzhou) and Peking University (PKU, in Beijing) are shown in Table S2. The similar size of these samples from different batches or labs demonstrated that the synthesis of LZU-111 and COF-300 are well-controlled and reproducible. Based on the crystals we synthesized, we have collected all the characterization and sorption data with narrow linewidth and symmetric peak shape in this manuscript. And all the characterization and sorption results can also be reproduced very well with corresponding COFs with specific crystal size synthesized in different batches. From this point, our comparative data support the results of controllable synthesis.

Table S1. Summary for controllable synthesis of different-sized LZU-111.

LZU-111							
Size	Aniline	Frozen	Rt Aging	Warming (40 °C)	Heating (120 °C)	Batch 1	Batch 2
~200 nm	—	liquid N ₂	0.5 h	—	3d		~1 μm ^a	15 eq, 0.07 mL	Liquid N ₂	1d	3d	3d		~30 μm	60 eq, 0.27 mL	Ice bath	3d	3d	4d		
a. The crystal size distributed from 700 nm to 1 μm, represented by ~1 μm-sized LZU-111.

Table S2. Summary for controllable synthesis of different-sized and COF-300.

COF-300 ^a							
Size	Aniline	Frozen	Rt Aging	Warming (40 °C)	Heating (120 °C)	Sample synthesized in Lanzhou University	Sample synthesized in Peking University
~500 nm ^b	–	Liquid N ₂	0.5 h	–	3d		~1 μm	0.6 eq, 5 μL	Liquid N ₂	1d	3d	3d		~30 μm	15 eq, 0.12 mL	Ice bath	3d	3d	4d		
a. The samples in Lanzhou University were synthesized in sealed glass tubes, and samples in Peking University were synthesized in sealed pressure tubes. *b.* These nanocrystals were synthesized with 15 M HOAc⁵.

The authors should provide statistical SEM information along with complementary characterization techniques such as DLS measurements to provide clear evidence that the given crystal sizes are representative. This is especially important considering that PXRD, solid state NMR, gas sorption and other characterizations requires a large quantity of samples to perform the test, and also because COF-300 500 nm and COF-300 1 μm have a very small size difference.

Response: Thank you very much for Reviewer’s valuable comments.

First, as per Reviewer’s suggestion, we have carefully analyzed the SEM images for each COF material in Fig. 2 and provided the statistical crystal size distribution as Fig. S4 in revised Supplementary Information.

Fig. S4 | Statistical crystal size distributions (corresponding to the SEM information in Fig. 2 in main text).

The statistics was implemented and analyzed with *Image J*. **a.** The size of nano LZU-111 crystal distributed from 150 to 270 nm and centered at 200 nm, represented by 200 nm-sized LZU-111. **b.** The size of LZU-111 crystal distributed mainly from 0.7 to 1.0 μm , represented by 1 μm -sized LZU-111. **c.** The size of micro-sized LZU-111 crystal distributed from 24 to 31 μm , represented by 30 μm -sized LZU-111. **d.** The size of nano COF-300 crystal mainly distributed from 450 to 600 nm and centered at 500 nm, represented by 500 nm-sized COF-300. **e.** The size of COF-300 crystal distributed mainly from 0.8 to 1.2 μm and centered at 1.0 μm , represented by 1 μm -sized COF-300. **f.** The size of micro-sized COF-300 crystal distributed from 24 to 33 μm , represented by 30 μm -sized COF-300.

Second, as suggested by Reviewer, DLS measurements is a technique which was used to determine the size distribution of small particles in suspension in solution. However, after many attempts, we found this technique is not very suitable for our COF samples for following reasons. **1)** From SEM images, we can see COF samples aggregated severely, even if the samples were treated after ultrasonication. **2)** If the samples were sonicated for long time and under high powder, parts of COF crystals were broken into pieces with different sizes (Fig. R1), which resulted in a polydispersity in the measurement. **3)** There were macroscopic sedimenting particles in the DLS measurements when crystal size is larger than 5 μm . **4)** COF-300 can be regarded as a material with swelling properties [*Science* 2018, 361, 48–52; *J. Am. Chem. Soc.* 2019, 141, 3298–3303], which is not suitable for DLS measurement. Specifically, with different guest molecules/solvents, the structure of flexible COF-300 transformed between contract and expanded phases. The unit cell parameter changed from $a = b = 19.6394(9)$ Å, $c = 8.9062(4)$ Å to $a = b = 26.2260(18)$ Å, $c = 7.5743(10)$ Å, with the unit-cell volume changed from 3435.2(4) Å³ to 5209.6(10) Å³. This nearly double increase in unit cell volume led to an obvious change on crystal size and morphology. As shown in Fig. R2, COF-300 crystals in water have rod-like morphology with crystal size of 55 $\mu\text{m} \times 10 \mu\text{m}$. After exchanged with 1,4-dioxane, the same batch of sample turned into bar-like morphology with crystal size of 45 $\mu\text{m} \times 15 \mu\text{m}$ [also see in our previous work: *Science* 2018, 361, 48–52]. Hence, there would be no comparison between crystal sizes measured in solvent (DLS) and those under vacuum (SEM). **5)**

Although water will not cause swelling of COF-300, it is not a good dispersant for organic hydrophobic COF material.

Fig. R1. Comparison of SEM images of 1 μm COF-300 crystals before and after sonication. It shows that after sonication over 15 mins, the crystals were broken into pieces with different sizes.

Fig. R2. Optical microscope photographs of COF-300 in water and in dioxane.

To clarify the representativeness of average crystal size of LZU-111 and COF-300, we provided more low-magnification SEM images of LZU-111 samples from different batches and COF-300 samples prepared in different labs in different cities (Lanzhou University and Peking University), which can help to provide more distribution information for samples, see in Table S1 and S2 (in Supplementary Information and also listed above). Besides, we added statistical SEM information of crystal size distribution (Fig. S4) as the supporting evidence. All these data showed that the synthesis of the LZU-111 and COF-300 crystals are well-reproduced, and each sample possesses high uniformity with very narrow size distribution.

Third, as pointed by reviewer, all the characterizations in our work require a large quantity of samples to perform the test. It is indeed impossible to get the narrow peaks and symmetric linetype of different characterizations if crystal size distributions were not uniform. For example, Xe is a very sensitive probe for porous materials. If crystal size or pore size distribution is not uniform, there will be different signals or chemical shift in ^{129}Xe NMR because of different mean free path [Zeolites 1988, 8, 350–361]. Therefore, all of our high-quality data revealed that the crystal size distribution is uniform.

Fourth, the crystal-size-controllable synthesis for COFs is very difficult. As shown in most of literatures of COFs, a lot of COFs have no well-defined morphology and crystal size. Although researchers have been working on controllable synthesis of COFs, but what kind of product can be obtained are serendipity in many cases. Not only that, even the size of crystallization domain is difficult to control. For example, similar modulated methods were reported to improve the crystallinity of crystallization domain [J. Am. Chem. Soc. 2014, 136, 8783-8789;

J. Am. Chem. Soc. 2015, 138, 1234–1239], but it is extremely difficult for crystallization domain to grow up from the nanoscale to the micron scale. Although 500 nm- and 1 μm -sized COF-300 have a very small size difference compared with 30 μm -sized samples, 500 nm-sized COF-300 is synthesized without modulator while 1 μm -sized COF-300 was synthesized by carefully adjusting the amount of modulator coordinated with various reaction conditions, which means even small differences can be finely tuned. Besides, as shown in our previous work and in this work (Fig. S5), COF-300 and LZU-111 of other crystal sizes can also be prepared by our method, which supports the controllability of our strategy.

Fig. S5 | SEM images of different-sized LZU-111 and COF-300 crystals. a–d, LZU-111 crystals with average sizes of $\sim 5 \mu\text{m}$, $\sim 10 \mu\text{m}$, $\sim 25 \mu\text{m}$, and $\sim 45 \mu\text{m}$, respectively. **e–f,** COF-300 crystals with average sizes of $\sim 4 \mu\text{m}$ and $\sim 10 \mu\text{m}$, which were used in the sorption experiments in Fig. S14.

Fifth, the characterization and sorption results can be reproduced very well with corresponding COFs with specific size in different batches. We would like to add more original sorption data of other sized samples (Fig. S14) produced at the time when we screened different controllable conditions to show that the crystal size effect influences the sorption behavior very regularly as we summarized without contingency. For example, as shown in Fig. S14, N_2 sorption experiments of $\sim 4 \mu\text{m}$ -sized and $\sim 10 \mu\text{m}$ -sized COF-300 were implemented and the isotherms were plotted and compared with those of other three sized COF-300. It shows that with the crystal size increased, the overall uptakes of N_2 are decreased very regularly although the overall uptake of 10 μm -sized

COF-300 is very close to that of 30 μm -sized COF-300. It is impossible to realize this if the synthesis cannot be reproduced or the crystal size distribution is dispersive.

Fig. S14 | N_2 adsorption-desorption isotherms of more different-sized COF-300 (solid circle, adsorption; hollow circle, desorption. 500 nm-sized, black; 1 μm -sized, red; 4 μm -sized, turquoise; 10 μm -sized, pink; 30 μm -sized, blue). The crystal size effect influences the sorption behavior very regularly as we summarized without contingency. As shown in Fig. S14, N_2 sorption experiments of $\sim 4 \mu\text{m}$ -sized and $\sim 10 \mu\text{m}$ -sized COF-300 were implemented and the isotherms were plotted and compared with those of other three sized COF-300 (500 nm-sized, 1 μm -sized, 30 μm -sized). It shows that with the crystal size increased, the overall uptakes of N_2 are decreased very regularly.

The authors intend to claim a different FMHW change trend between LZU-111 and COF-300. Indeed, a noticeable FMHW difference is seen between COF-300 500 nm and COF-300 1 μm in Figure 3, inconsistent with the claim of “nearly the same” (Page 7). The authors should give the FMHW values for each COF crystal. In addition, the authors investigated the crystal size effect on PXRD with LZU-111 crystals (200 nm, 1 μm , 30 μm) and COF-300 crystals (500 nm, 1 μm , 30 μm). The crystal sizes found in polycrystalline COF powder may vary substantially, which should also be considered. From Figure 3 a and 3c, an alternative understanding could be that FMHW shows a large difference with the crystal size below 1 μm , while a minor difference with the crystals larger than 1 μm . The authors should clarify this.

Response: According to the Reviewer’s helpful suggestion, we have made following revisions.

- 1) The FWHM values of the first peak at 8.9° in PXRD patterns of 500 nm-sized, 1 μm -sized and 30 μm -sized COF-300 are 0.158° , 0.158° , 0.122° , respectively. Compared with the FWHM values of different-sized LZU-111 (all $> 0.2^\circ$), all these peaks of COF-300 are very narrow, indicating very high crystallinity of

different-sized COF-300. What's more, for 500 nm-sized crystals, the sample particles were used to collect single-crystal 3D rotation electron diffraction (RED) data [*J. Am. Chem. Soc.* 2018, *140*, 22, 6763–6766], which confirmed the mono-crystallinity of the small crystals.

For more accuracy, we revised this sentence in Paragraph 5 on Page 7 as: “... the FWHM values of the (200) peak at 8.9° in PXRD patterns of 500 nm-, 1 μm- and 30 μm-sized COF-300 are 0.158°, 0.158°, 0.122°, respectively (Fig. 3c). Although the FWHM is further decreased a little in 30 μm-sized sample, it should be noted that all different-sized COF-300 crystals keep their single-crystallinity, since the small and large COF-300 crystals were suitable for either single-crystal electron diffraction or SXRD structural analysis.^{27,28}”

- 2) We fully agree with Reviewer's comment about the impact of crystal size on FWHM. Generally speaking, when the crystal size is greater than 5 μm, no significant difference in FWHM could be distinguished by PXRD if samples have similar crystallinity. For LZU-111, although the FWHM values of PXRD peaks indeed increased from 0.338° to 0.221° to 0.211° with the increasing crystal size, the FWHM difference between 1 μm and 30 μm (0.117°) is smaller than that between 200 nm and 1 μm (0.010°). Besides providing specific values, we would like to add more detailed explanation in Paragraph 4 on Page 6 as kindly suggested by Reviewer: “(e.g., FWHM values of the strongest reflection peak at 5.6° are 0.338°, 0.221°, 0.211° for 200 nm-, 1 μm- and 30 μm-sized LZU-111, respectively, Fig. 3a). It shows a large FWHM difference between 200 nm- and 1 μm-sized crystals (0.117°), while a minor FWHM difference can be observed between 1 μm- and 30 μm-sized LZU-111 crystals (0.010°).” We also added this information in the analyses of corresponding sorption experiments and ¹²⁹Xe NMR experiment in revised main text and Supplementary Information.

Paragraph 6 on page 10 in main text: “Note that BET surface area of 1 μm-sized LZU-111 is calculated to be 1870 m² g⁻¹, which is much larger than that of 200 nm-sized sample and relatively closer to that of 30 μm-sized sample. This is in good agreement with the trend of crystallinity change along with crystal size (Fig. 3a).”

Fig. S12 in Supplementary Information, pore size distributions: “This is also in good agreement with the trends of crystallinity change (Fig. 3a) and BET surface area change (Fig. 4a).”

Fig. S20 in Supplementary Information: “The intervening FWHM of ~1640 Hz is observed for the signal of 1 μm-sized LZU-111.” and “Note that chemical shifts for 1 μm- and 30 μm-sized LZU-111 are very similar (~77 ppm) and largely different from that of 200 nm-sized LZU-111 (~155 ppm), which matched very well with the crystallinity change trends observed from PXRD (Fig. 3a).”

- 3) The description of line width of NMR signals for different samples were added in Paragraph 4 on Page 7 as: “For example, the line widths of signal at 64 ppm assigned to the quaternary carbon atom in TAM linker are 165 Hz, 85 Hz, 69 Hz for 200 nm-, 1 μm- and 30 μm-sized LZU-111, respectively (Fig. 3b).” And in Paragraph 5 on Page 8 as: “There is a slight change regarding the linewidth of SSNMR signals in different sized COF-300. For example, signal at 65 ppm assigned to the quaternary carbon atom in TAM linker for 500 nm-, 1 μm- and 30 μm-sized COF-300 are 94 Hz, 72 Hz, 59 Hz, respectively.”

COF-300 1 μm shows extra peaks at around 11°, 13° and 15°, which are absent for COF-300 500 nm and COF-300 30 μm. What is the reason for that?

Response: Thank you very much for Reviewer's question. This sample might be accidentally contaminated in the process of filling capillary to collect high-resolution PXRD. Before the sorption experiments, we had checked the PXRD of 1 μm -sized COF-300 samples to confirm its phase purity and did not find any extra peaks that time (Fig. R3, data was collected with different diffractometer which results in different peak width). Recently we recollected a new PXRD data for 1 μm -sized COF-300 sample (Fig. R4) with the same diffractometer which was used to collect all datasets in Fig. 3c, and also found there are no extra peaks in Fig. R4. We replaced this pattern with the recollected one in Fig. 3c in the revised manuscript (also shown here).

Fig. R3. PXRD pattern of 1 μm -sized COF-300 samples before sorption experiments. The data was collected on Rigaku D/Max-2400 diffractometer with step size of 0.02° , exposure time of 0.2 s.

Fig. R4. Recollected PXRD pattern of 1 μm -sized COF-300 samples with PANalytic high-resolution diffractometer which we used to collect all datasets in Fig. 3c under the same conditions. This pattern was presented in revised Fig. 3c as the red line.

Fig. 3 | PXRD and SSNMR for different-sized LZU-111 and COF-300. a, PXRD patterns of different-sized LZU-111. Inset: magnified patterns of $2\theta = 6 - 35^\circ$ with normalized intensity. **b,** ^{13}C CP/MAS spectra of different-sized LZU-111. The assignments of ^{13}C chemical shifts are indicated in the chemical structure. **c,** PXRD patterns of different-sized COF-300. **d,** ^{13}C CP/MAS spectra of different-sized COF-300. Asterisks denote spinning sidebands. The assignments of ^{13}C chemical shifts are indicated in the chemical structure.

COF-300 exhibits significant differences in the solid state NMR spectra with varying crystal size, which is a very interesting result. However, it is conceivable that the differences in the ^{13}C NMR spectra are due to varying amounts of residual modulator (or changes in hydration/solvation of the pore system, which has been reported to be a common phenomenon in this system). This is what the authors themselves allude to in the SI, p. 23: “The possible reason is, comparing with 500 nm-sized crystals synthesized without modulator, 1 μm -sized crystals synthesized with aniline has a considerable number of different imine bonds from aniline reacting with aldehyde linkers at the edges of crystals.” without discussing this issue further and its impact on the observed changes in sorption behavior. Unless the origin of the different signals are unambiguously clarified, no meaningful conclusion regarding crystal size effects can be drawn from these data. In Figure S5, the authors claimed the signals of $-\text{CHO}$ and $-\text{NH}_2$ end groups could be observed for the 200 nm-sized crystals but they are absent for

the 30 μm -sized crystals, and related the weaker $-\text{CHO}$ and $-\text{NH}_2$ peak in LZU-111 30 μm with less defects and increased crystallinity. However, LZU-111 1 μm and LZU-111 30 μm show a noticeable intensity at around 114 ppm, which may be due to $-\text{NH}_2$. The authors should provide more NMR spectral evidence to clarify this. Moreover, the fact that the CHO carbon signal is observed in LZU-111 200 nm might be that aniline was not used in the reaction and CHO remained as end group. Instead, a large amount of aniline was used for the synthesis of LZU-111 1 μm and LZU-111 30 μm , which could result in the disappearance of CHO end group carbon signal.

Response: We appreciate the reviewer's positive comments expressed for our interesting result, and also thank for these interesting questions regarding the residual modulator, terminal groups/defects and hydration/solvation of the pore system in different-sized samples.

- 1) As discussed in our previous work, the role of the mono-functional modulator aniline is to suspend fast nucleation and increase the reversibility of the whole crystal growth process. Although aniline was used to react with aldehyde linker first, it was replaced by amine linker to form COF in the following process by reversible imine exchange reaction. For large 30 μm -sized single-crystal COFs, it would not be possible to form connected framework of large single crystals if many monofunctional modulator molecules exist and intercept the connection. And it has already been proved by atomically accurate SXRD technique that residual **aniline can hardly be detected in single-crystal COFs** [*Science* 2018, 361, 48–52].
- 2) **Nano-sized COFs contain no modulator** since they were synthesized without modulator.
- 3) For 1 μm -sized crystals, considering aniline was used in the synthetic process, and the crystal size is not as large as 30 μm -sized crystals, we inferred that 1 μm -sized crystals contain relatively more residual modulators compared with 30 μm -sized crystals. This might be responsible for several special phenomena in sorption experiments with very sensitive guests or those guests who have strong interactions with COFs (e.g., a slight slope change in alcohols absorption on 1 μm -sized crystals). However, **the amount of these residual modulators is very little**. To prove this point, we provided new mass spectrometry (MS) analyses data for digested 1 μm -sized crystals (Fig. R5) and pure aniline (Fig. R6). As shown in Fig. R5, there is no signal of aniline (+MS ~93.7, Fig. R6) can be probed for digested 1 μm -sized crystals by MS, indicating the aniline in 1 μm -sized LZU-111 was at least below the detection limit. And it is readily comprehensible that, in this reversible reaction, not all the terminal groups in the material have to react with aniline—the unreacted $-\text{CHO}$ and aniline could exist in the meantime. Hence, it can help to rule out the possibility of the existence of a large amount of aniline, **and these little residuals cannot be detected by SSNMR**.
- 4) Regarding the hydration, the whole hydrophobic organic framework of COFs greatly reduces the possibility of strongly adsorbing water firstly. To avoid the possible influence of hydration/solvation of the pore system on the sorption results, we carefully carried out our experiments following the steps as shown below. Our samples were first activated as following procedures: Soxhlet extraction in 1,4-dioxane for 24 h, drying at ambient temperature for 12 h, at 100 °C for 12 h, and at 120 °C in vacuum for 12 h to 24 h. Then the sorption experiments were carried after sample activation under 120 °C for 12 h on the sorption instrument and continuously starting adsorption, without exposing samples to air/external environment. Before the NMR experiments, the samples were re-activated to rule out the possibility that samples may adsorb water from the air. In the measurements, the samples were sealed in a zirconium rotor with a cap on it. Note that our NMR experiments were carried at the lab in Lanzhou which has a dry temperate continental climate, reducing the possibility of water absorption from air. Hence, **there are no hydration signals can be detected in SSNMR of COFs**.

- 5) We really agree with the Reviewer's opinion that the residual –CHO could react with aniline, however, not all –CHO in the material have to react with aniline in the reversible reaction—it means that the unreacted terminal –CHO and aniline could exist in the meantime. It is supported by the FT-IR data that there is still a minor signal of –CHO in 1 μm -sized LZU-111 crystals because of higher detection sensitivity of FT-IR than that of SSNMR. Considering the Reviewer's comment, two factors were summarized as the reasons that the minor signals of –CHO disappeared in 1 μm - and 30 μm -sized LZU-111 crystals in ^{13}C CP/MAS spectra: One could be that, part of the residual –CHO in 1 μm - and 30 μm -sized crystals can react with aniline. Another one is the increasing of crystallinity with enlarged crystal size. When the crystal size is large enough or the crystallization is good enough, the condensation reaction occurs more completely. With the increasing of the crystal size, larger single crystals contain relatively less external surface area and terminal groups than small crystals. For example, from SSNMR spectrum of 500 nm-sized COF-300, there is no signal of terminal –CHO. It means that the signal of terminal group cannot be observed in SSNMR when COFs have high crystallinity. Accordingly, we revised this description on Page P16 in Supplementary Information: "...the minor signals at 190 (corresponding to the terminal –CHO group) and 114 ppm (related to the terminal –NH₂ group) are the indicators for evaluating the possible defects in LZU-111. **These signals could be obviously observed for the 200 nm-sized crystals but are nearly absent for the 30 μm -sized crystals, first because with the increasing of the crystal size, the condensation reaction occurs more completely, thus pore integrity of LZU-111 is significantly improved along with the increased crystal size and crystallinity. Besides, part of the residual –CHO in 1 μm - and 30 μm -sized crystals could react with aniline in reversible imine condensation/exchange reactions.**"
- 6) The reviewer mentioned that there is "noticeable signal" at ~115 (114.7) ppm in SSNMR spectra of 1 μm - and 30 μm -sized LZU-111. Although there might be very minor terminal –NH₂ groups according to the reaction reversibility, the magnified ^{13}C /CP MAS spectra range from 100 to 170 ppm (Fig. R7) show that there is no obvious signal around 115 ppm. We revised this description as follows for more accuracy: "**These signals could be *obviously* observed for the 200 nm-sized crystals but are *nearly* absent for the 30 μm -sized crystals, ...**"
- 7) As per Reviewer's suggestion, and considering all the analyses above, the assignments of chemical shifts were added in the NMR spectra (Fig. 3b and 3d).
- 8) We are grateful for receiving these precious questions, which gave us very good opportunity to reexamine our ideas, our data and all the details. In this work, the growing of crystal size was visualized, all the different data supported well with each other, and all the results especially the variation tendency along with change of crystal size are in very good agreement, which means the important crystal size effects in COFs are presented very well based on the realization of crystal-size-controlled synthesis, and they are worth studying. We understand that all the materials inevitably have terminals and defects, but unless there are a large amount terminals and defects which were produced by doping or there are very strong/sensitive host-guest interactions, it would not influence the main properties of materials.

Fig. R5. MS analysis for digested 1 μm -sized LZU-111 crystals. It shows that there is no signal of aniline. m/z calcd. for $\text{C}_{25}\text{H}_{24}\text{N}_4$ (TAM) $[\text{M} + \text{H}]^+$: 380.48, found: 381.1507.

Fig. R6. MS analysis for aniline. MS: m/z calcd. for C_6H_7N (aniline) $[M + H]^+$: 93.1, found: 93.7.

Fig. R7. Magnified ^{13}C CP/MAS spectra of three different-sized LZU-111 range from 100 to 170 ppm. It shows that there is no obvious signal around 115 ppm in the spectra of 1 μm - and 30 μm -sized LZU-111.

Did the author try crystal-size controlled synthesis for LZU-79, which indeed gave the largest crystal in their previous paper (Ref 27)? As LZU-79 should adopt similar configuration as COF-300, investigating LZU-79 crystal size effect on gas sorption could further verify the generality of the relationship of pore structure and crystal size.

Response: We also tried crystal-size controlled synthesis for LZU-79 and the related project is under the way. The very interesting thing is, although LZU-79 adopts a similar topology with COF-300 (**dia**-based), it is quite unique because it contains an imidazole group as an active functional group at the side chain (not minor terminal groups but abundant and homogeneously distributed in the framework), resulting in a very different host-guest interaction with guest molecules in the pores. Such kind of functional group will lead to different sorption behavior. Based on these, we think LZU-79 is a very interesting materials, whose sorption behavior could be classified into another type, which is different from the sorption behaviors of COF-300 and LZU-111 which have no more additional active group at the side chain.

Other comments:

p. 6: “ the reflection intensity gradually increases along with the increasing crystal size”: How was the XRD intensity quantified? Unless the amount of material in the X-ray beam is precisely known or an internal standard is used, comparison of X-ray intensities gives qualitative trends at best.

Response: We fully agree with the Reviewer that the most accurate and strict quantification method in PXRD is using an internal standard. Here, we did not strictly quantify them as specific value but defined as a variation tendency. To make sure this variation tendency is reasonable, in our experiment, the same amount of different-sized LZU-111 samples was used to collect PXRD data with the same diffractometer and same parameter setting, and under the constant environment temperature. This variation tendency is well-reproducible because the

crystallinity is indeed increased with crystal size growing up. But for the scientific accuracy, we deleted the discussion about comparing the intensity and normalized all the PXRD data for comparison on Page 7 in the revised manuscript.

p. 7: It is not clear what the authors mean with the term “structure anisotropy”. Please clarify.

Response: As kindly suggested by Reviewer, we added the definition of “structure anisotropy” by revising this sentence as “...which can be ascribed to the typically enhanced structure anisotropy of larger crystals in NMR crystallography,³¹ **resulting in two chemically same atoms show different chemical shifts in SSNMR due to the crystallographic inequivalence.**”

In Figure 3b, 3d and S5, the denotation of asterisks should be described in the figure caption.

Response: Thank you very much for Reviewer’s suggestion and this helps to clarify this part. According to the reviewer’s suggestion, we added the description of the denotation of asterisks as “**Asterisks denote spinning sidebands**” in the caption of Figures 3d and S6.

Fig S17: “The blue isotherm matches well with the red isotherm, which means that 30 μm-sized single crystals of COF-300 keeps its contracted phase during N₂ adsorption”. The agreement between the experimental and calculated isotherms is limited, which probably is due to the fact that contraction/expansion of the COF-300 is a dynamic process which is hard to model. This discrepancy is even more obvious for the black/turquoise isotherms. I therefore encourage the authors to model the sorption behavior by more accurate methods than classical DFT, i.e. using atomistic calculations.

Response: We thank the Reviewer’s suggestion about using atomistic calculations. It is worth noting that there is no reliable calculation method for predicting or simulating sorption isotherm of flexible material by considering the crystal size effect. This is the reason why we focused on the structure of stable intermediate state and the final state, i.e., calculating the overall adsorbing capacity of contraction/expansion structure to prove that the pores can be expanded with N₂ in nano COF-300 crystals while the pores in large single crystals can keep contracted under N₂. Our calculation results in Fig. S19 show that the experimental adsorbing capacity of large single crystals (red) is approximate to the calculated adsorbing capacity of contracted structure (blue), indicating that the contracted structure of large single crystals cannot be enlarged during N₂ sorption. On the contrast, the experimental adsorbing capacity of small-sized nano crystals (black) is approximate to the calculated adsorbing capacity of expanded structure (turquoise), indicating that the initial contracted structure of small-sized nano crystals can be expanded to the final expansion state after filling with N₂. Although the dynamic structure change process of COF-300 upon gas adsorption was not involved in this manuscript, the method for the calculating dynamic structure change upon gas adsorption with considering the crystal size effect need to be further developed in the following work.

Fig 4: What is the rationale behind the fact that the hysteresis of the smallest sized COF-300 crystals is typically the largest among the different sizes?

Response: Thank you for Reviewer’s comment. This is a very interesting question. The size and shape of hysteresis loop are generally considered to be related to many complicated factors such as pore properties (e.g., shape of pore structure, pore openness, surface roughness, etc.), different host-guest interactions, and even measurement settings (e.g., equilibration time) in sorption experiment. The hysteresis loops with different sizes

and shapes in the sorption processes of flexible COF-300 strongly rely on the interactions between the different guests (gases/vapors) and the different-sized host (COF-300 crystals). These different interactions determine when and how the COF-300 crystals expand or shrink (also see the summary of relationship between gases/vapors properties and sorption behaviors of different sized COF-300 in Table S3). For example, the smallest sized COF-300 (500 nm) has the largest hysteresis in N₂/Ar sorption because it has the largest overall uptake, owing to the most flexible framework with least structural unit.

To better understand this question, we calculated the areas of hysteresis loops of CO₂ adsorption-desorption isotherms (the areas of closed curves in OriginLab), since the shapes of hysteresis loops of CO₂ are not very regular. Interestingly, although it looks the smallest COF-300 has the largest hysteresis, the results shows that the absolute value of hysteresis areas in isotherms of 500 nm-, 1 μm-, and 30 μm-sized COF-300 are 226.6, 335.6, 346.0, respectively, see Fig. R8-R10. It is worth mentioning that we have been working on this project to study the structure of each adsorption stage by using *in-situ* Sorption-SXRD analysis, which will be presented in the other works.

Table S3. Relationship between gases/vapors properties and sorption behaviors of different sized COF-300.

Gas/ vapor	Molecular size and shape	Polarity comparison	Gate-open pressure (P/P_0)			Sorption capacity ($\text{cm}^3_{\text{STP}} \text{g}^{-1}$)		
			500 nm	1 μm	30 μm	500 nm	1 μm	30 μm
Ar 	sphere	Ar < N ₂ < CO ₂	0.05	0.025 ^a	NA ^b	364.4	206.8	122.5
N ₂ 	2.99 Å ellipsoid		0.05	0.028 ^a	NA ^b	436.9	274.4	115.2
CO ₂ 	linear		0.49	0.149	0.129	372.5	423.9	426.4
Tetrahydrofuran 	4.053 Å ring molecule	THF < 1,4-dioxane < isopropanol < ethanol	0.038	0.039	0.015	179.0	181.8	180.4
1,4-dioxane 	4.796 Å ring molecule		0.055	0.062	0.027	171.7	174.2	171.9
Isopropanol 	4.322 Å rod-like		0.011	0.091	0.085	178.2	179.3	180.5
Ethanol 	4.784 Å rod-like		0.273	0.276	0.256	219.5	221.1	223.5

a. The pores were partially opened in 1 μm -sized COF-300 with N₂ and Ar. *b.* The pores can hardly be opened in 30 μm -sized COF-300 with N₂ and Ar, see Fig. S19.

Fig. R8. The adsorption-desorption isotherms of 500 nm-sized COF-300 and the corresponding calculation results for the area of hysteresis.

Fig. R9. The adsorption-desorption isotherms of 1 μm -sized COF-300 and the corresponding calculation results for the area of hysteresis.

Fig. R10. The adsorption-desorption isotherms of 30 μm -sized COF-300 and the corresponding calculation results for the area of hysteresis.

The authors should also provide ^{129}Xe NMR measurements for COF-300 and the data should be discussed in the context of the crystal size effects invoked in this manuscript.

Response: At the suggestion of the Reviewer, we added the ^{129}Xe NMR spectra of xenon adsorbed within the different-sized COF-300 as Fig. S21 in Supplementary Information (also shown here) and discussed it in detail. We also added the related description in the revised main text on Page 11 as: “**Besides, the ^{129}Xe NMR**

measurements confirmed that the extents of pore expansion with filling of these inert gases in various-sized COF-300 are very different (Fig. S21).”

Fig. S21| ^{129}Xe NMR spectra of xenon adsorbed within the different-sized COF-300 (Xe@500 nm-sized COF-300, black; Xe@1 μm -sized COF-300, red; Xe@30 μm -sized COF-300, blue). The chemical shifts of ^{129}Xe NMR signals increase from 152 to 176 to 189 ppm along with the increasing crystal size of COF-300, indicating that the enlarged pore size is decreased^{7,8} along with the increasing crystal size. The symmetric peak shape of 500 nm-sized COF-300 implies that the pores in nano-sized COF were totally opened to uniform expanded pore, while the unsymmetrical peak shape of micro-sized COF-300 suggests that pores in these samples were partially expanded, resulting in heterogeneous pore size. Besides, as one of inert gases, the overall uptake of Xe in different sized COF-300 decrease along with the increasing crystal size because of enhanced framework rigidity, similar with sorption behaviors of N_2 and Ar in different-sized COF-300 (Figs. 4c-4d). Specifically, in the process of sample preparation of Xe adsorbing into samples before NMR measurement, the highest sorption pressure for 30 μm -sized COF-300 is 5 mbar, while for 500 nm-sized and 1 μm -sized COF-300 are both 10 mbar. Then the three-sized Xe@COF-300 were used to collect ^{129}Xe NMR spectra. As shown in Fig. S21, there is a very high free state/gas Xe signal of 0 ppm in the spectrum of 30 μm -sized COF-300, which means that less than 5 mbar Xe can be adsorbed in the most rigid 30 μm -sized COF-300. In other word, 30 μm -sized COF-300 only absorbed part of 5 mbar Xe in the pore and the rest of Xe is still as free state/gas Xe. There are both free state/gas Xe signal of 0 ppm and adsorbed state Xe signal of 176 ppm in the spectrum of 1 μm -sized COF-300, which means that less than 10 mbar Xe can be adsorbed in 1 μm -sized COF-300. The only one adsorbed state Xe signal of 152 ppm in the spectrum of 500 nm-sized COF-300 suggests that, 500 nm-sized COF-300 absorbed all 10 mbar Xe as adsorbed state Xe. All these results verify again about our conclusion that the inert gases can hardly open the framework of 30 μm -sized COF-300 because crystal size controls structural flexibility of COF-300 by altering the number of repeating units, which eventually changes sorption selectivity.

REVIEWERS' COMMENTS

Reviewer #1 (Remarks to the Author):

As can be seen from the revised manuscript and the carefully prepared point-by-point response letter, major issues on SSNMR, PXRD, guest related size-flexibility relationship, and residual modulators have been addressed. The corrections in current version are satisfactory to increasing the quality of paper. Therefore, the revised manuscript could be suitable for publication in Nat. Commun.

Reviewer #2 (Remarks to the Author):

I am satisfied with the changes made and this manuscript could be accepted as it is now.

Reviewer #4 (Remarks to the Author):

Crystal size effect is indeed very important in materials science, as it can significantly influence the properties and functions of materials. As an emerging class of porous materials, COFs have gained intensive attentions in the past decade. However, the crystal size effect of COFs on functionality has never been investigated. In this manuscript, the authors reported the first example on crystal-size controlled synthesis of COFs. By taking LZU-111 and COF-300 as examples, they clearly demonstrate the mesoscopic control of properties and functions in COFs with different crystal sizes. Personally, I think this is a really nice result and the characterization is good. Moreover, I think this result will push the COF field for practical applications. Therefore, I would like to recommend the publication.

Point-by-Point Response to Reviewers' Comments

The comments of each reviewer are copied here in their entirety (*italics*) and our responses are given below for each segment of comments.

Reviewer #1 (Remarks to the Author):

As can be seen from the revised manuscript and the carefully prepared point-by-point response letter, major issues on SSNMR, PXRD, guest related size-flexibility relationship, and residual modulators have been addressed. The corrections in current version are satisfactory to increasing the quality of paper. Therefore, the revised manuscript could be suitable for publication in Nat. Commun.

Response: We deeply appreciate the Reviewer's valuable comments and the recommendation for our work.

Reviewer #2 (Remarks to the Author):

I am satisfied with the changes made and this manuscript could be accepted as it is now.

Response: We sincerely appreciate the Reviewer's valuable comments and recommendation.

Reviewer #4 (Remarks to the Author):

Crystal size effect is indeed very important in materials science, as it can significantly influence the properties and functions of materials. As an emerging class of porous materials, COFs have gained intensive attentions in the past decade. However, the crystal size effect of COFs on functionality has never been investigated. In this manuscript, the authors reported the first example on crystal-size controlled synthesis of COFs. By taking LZU-111 and COF-300 as examples, they clearly demonstrate the mesoscopic control of properties and functions in COFs with different crystal sizes. Personally, I think this is a really nice result and the characterization is good. Moreover, I think this result will push the COF field for practical applications. Therefore, I would like to recommend the publication.

Response: We sincerely thank the Reviewer for the valuable comments and strong

support expressed for our work.